EMBO
Molecular Medicine

# Loss of Mpdz impairs ependymal cell integrity leading to perinatal-onset hydrocephalus in mice

Anja Feldner[1], M Gordian Adam[1,†], Fabian Tetzlaff[1], Iris Moll[1], Dorde Komljenovic[2], Felix Sahm[3,4], Tobias Bäuerle[2,‡], Hiroshi Ishikawa[5], Horst Schroten[6], Thomas Korff[7], Ilse Hofmann[8,9], Hartwig Wolburg[10], Andreas von Deimling[3,4] & Andreas Fischer[1,9,11,*] [ID]

## Abstract

Hydrocephalus is a common congenital anomaly. *LCAM1* and *MPDZ* (*MUPP1*) are the only known human gene loci associated with non-syndromic hydrocephalus. To investigate functions of the tight junction-associated protein Mpdz, we generated mouse models. Global *Mpdz* gene deletion or conditional inactivation in Nestin-positive cells led to formation of supratentorial hydrocephalus in the early postnatal period. Blood vessels, epithelial cells of the choroid plexus, and cilia on ependymal cells, which line the ventricular system, remained morphologically intact in *Mpdz*-deficient brains. However, flow of cerebrospinal fluid through the cerebral aqueduct was blocked from postnatal day 3 onward. Silencing of *Mpdz* expression in cultured epithelial cells impaired barrier integrity, and loss of *Mpdz* in astrocytes increased RhoA activity. In *Mpdz*-deficient mice, ependymal cells had morphologically normal tight junctions, but expression of the interacting planar cell polarity protein Pals1 was diminished and barrier integrity got progressively lost. Ependymal denudation was accompanied by reactive astrogliosis leading to aqueductal stenosis. This work provides a relevant hydrocephalus mouse model and demonstrates that *Mpdz* is essential to maintain integrity of the ependyma.

**Keywords** aqueductal stenosis; cerebrospinal fluid; ependymal cells; hydrocephalus; tight junction
**Subject Category** Neuroscience

## Introduction

Congenital hydrocephalus, an abnormal accumulation of cerebrospinal fluid (CSF) in brain cavities, is diagnosed in ~1 of 2,000 newborns (Schrander-Stumpel & Fryns, 1998; Garne *et al*, 2010). Increasing amounts of CSF in the ventricular system raise the intracranial pressure leading to compression of brain tissue and enlargement of the head circumference. The most common treatment is surgical insertion of a catheter to drain excess CSF into another body cavity. However, this does not reverse 80–90% of the neurological impairment of neonates with fetal onset hydrocephalus. Therefore, hydrocephalus may not only be a disorder of CSF dynamics, but also a brain disorder (Guerra *et al*, 2015).

Disturbance of CSF formation, flow, or absorption can cause congenital hydrocephalus (Jiménez *et al*, 2014). Overproduction of CSF has been described in rare cases, in particular choroid plexus hyperplasia or papilloma (Fujimoto *et al*, 2004). Brain malformations (e.g., Arnold–Chiari or Dandy–Walker syndrome) can obstruct CSF flow (Schrander-Stumpel & Fryns, 1998), which is driven by pressure gradients generated, for example, by blood pulsations. However, it is assumed that motile, water-propelling cilia on ependymal cells lining the ventricular system are also required for proper CSF flow (Lee, 2013; Jiménez *et al*, 2014). This hypothesis is supported by the fact that some ciliopathies are associated with hydrocephalus and that various rodent models carrying mutations in genes required for cilia function develop postnatal hydrocephalus (Davy & Robinson, 2003; Ibañez-Tallon *et al*, 2004; Banizs *et al*, 2005; Lechtreck *et al*, 2008; Jacquet *et al*, 2009; Wodarczyk *et al*, 2009; Tissir *et al*, 2010; Liu *et al*, 2014; Koschützke *et al*, 2015;

1 Vascular Signaling and Cancer, German Cancer Research Center (DKFZ), Heidelberg, Germany
2 Division of Medical Physics in Radiology, German Cancer Research Center (DKFZ), Heidelberg, Germany
3 Department of Neuropathology, Institute of Pathology, Ruprecht-Karls-University Heidelberg, Heidelberg, Germany
4 Clinical Cooperation Unit Neuropathology, German Consortium for Translational Cancer Research (DKTK), German Cancer Research Center (DKFZ), Heidelberg, Germany
5 Department of NDU Life Sciences, School of Life Dentistry, Nippon Dental University, Chiyoda-ku, Tokyo, Japan
6 Pediatric Infectious Diseases, University Children's Hospital Mannheim, Heidelberg University, Mannheim, Germany
7 Department of Cardiovascular Research, Institute of Physiology and Pathophysiology, Heidelberg University, Heidelberg, Germany
8 Vascular Oncology and Metastasis, German Cancer Research Center (DKFZ), Heidelberg, Germany
9 Vascular Biology, CBTM, Medical Faculty Mannheim, Heidelberg University, Mannheim, Germany
10 Department of Pathology and Neuropathology, University of Tuebingen, Tuebingen, Germany
11 Medical Clinic I, Endocrinology and Clinical Chemistry, Heidelberg University Hospital, Heidelberg, Germany
*Corresponding author. Tel: +49 6221 424150; E-mail: a.fischer@dkfz.de
†Present address: Immunocore Limited, Abingdon, Oxon, UK
‡Present address: Institute of Radiology, University Medical Center Erlangen, Friedrich-Alexander-Universität Erlangen-Nürnberg, Erlangen, Germany

 

Rachel *et al*, 2015). Lastly, congenital hydrocephalus can also be acquired due to prenatal cerebral infections, injuries, or hemorrhages, which either inhibit CSF flow or CSF reabsorption (Tully & Dobyns, 2014).

A genetic etiology is assumed for a large proportion of patients suffering from congenital hydrocephalus. Up to date, mutations that cause non-syndromic congenital hydrocephalus in humans have been detected in only two genes. (i) Mutations in *L1CAM* on chromosome Xq28 are responsible for X-linked recessive congenital hydrocephalus (HSAS, OMIM #307000). *L1CAM* mutations can also lead to syndromic disorders (L1 syndrome, CRASH syndrome) associated with hydrocephalus (Tully & Dobyns, 2014). (ii) Homozygous *MPDZ* loss-of-function mutations on chromosome 9p23 are responsible for autosomal recessive non-syndromic hydrocephalus (HYC2, OMIM #615219). Fetuses carrying truncating *MPDZ* mutations developed macrocephaly, extreme dilation of the lateral ventricles with dangling of the choroids and thinning of the cerebral cortex (Al-Dosari *et al*, 2013).

Interestingly, both L1CAM and the multi-PDZ domain protein (MPDZ, also known as MUPP1) are involved in cell–cell adhesion. In mice, loss of the *junctional adhesion molecule C* (*Jam3*) also causes hydrocephalus (Wyss *et al*, 2012). It was therefore hypothesized that disruption of the junctions between cells of the ventricular zone may be the common cause (Al-Dosari *et al*, 2013; Jiménez *et al*, 2014; Guerra *et al*, 2015). Mechanistically, this could occur due to reactive astrogliosis following ependymal cell injury (Wagner *et al*, 2003) to re-establish a surface between brain tissue and CSF (Sarnat, 1995). In particular, astrogliosis within the cerebral aqueduct would then obstruct CSF flow.

MPDZ contains 13 PDZ domains, which mediate multiple protein–protein interactions, and thereby acts as a scaffold protein for tight junction-associated proteins (JAM-A, claudin-1, ZO-3, Pals1, Par6), adherens junction proteins (Nectins), transmembrane receptors (Adachi *et al*, 2009), and the RhoA-specific guanine exchange factor Syx (PLEKHG) (Estévez *et al*, 2008). *Mpdz* is expressed ubiquitously in the brain of mouse embryos with highest mRNA levels in the ependymal lining of the ventricular system, and the choroid plexus (Ullmer *et al*, 1998; Bécamel *et al*, 2001; Sitek *et al*, 2003). The role of MPDZ for restricting epithelial permeability is unclear. Silencing of *MPDZ* expression did not alter paracellular permeability of EpH4 breast carcinoma cells (Adachi *et al*, 2009). However, loss of MPDZ may be compensated by increased expression of INADL (also known as InaD-like or PATJ), a structural paralogue of the MPDZ protein (Adachi *et al*, 2009; Assémat *et al*, 2013).

In this study, we generated global and conditional mouse models to elucidate the pathogenesis of autosomal recessive non-syndromic hydrocephalus by targeting the murine *Mdpz* locus.

# Results

### Targeting *Mpdz* in mice

A homozygous *MPDZ* nonsense mutation in exon 6 which truncates the MPDZ protein after the first PDZ domain was found in patients suffering from autosomal recessive hydrocephalus (Al-Dosari *et al*, 2013). To target *Mpdz* in mice, we used embryonic stem cells, in

which a stop cassette had been inserted into intron 11–12 (clone XG734; BayGenomics), and injected into mouse blastocysts. This strategy leads to a premature stop codon and truncates the Mpdz protein after the 3$^{rd}$ PDZ domain (Fig 1A). After germline transmission, we obtained heterozygous *Mpdz*$^{Gt(XG734)Byg(+/-)1AFis}$ mice (*Mpdz*$^{+/-}$ mice), which were viable and fertile, and backcrossed into the C57BL/6 strain. Western blotting revealed that intact Mpdz protein could no longer be detected in brain lysates derived from neonatal *Mpdz*$^{-/-}$ mice (Fig 1B).

In a second approach, loxP sequences were inserted into introns 3–4 and 5–6 by homologous recombination in ES cells. This allows recombination of the floxed allele by Cre recombinase to delete exons 4 and 5 (Mpdz$^{Δ}$) resulting in a truncating frameshift mutation removing all 13 PDZ domains (Fig 1C). Cre recombinase was expressed under control of the ubiquitously active CMV promoter (Schwenk *et al*, 1995), leading to a strong reduction of Mpdz protein expression levels in astrocytes isolated from neonatal mice (Fig 1D) Homozygous floxed *Mpdz*$^{fl/fl}$ mice and heterozygous CMV-Cre;*Mpdz*$^{+/Δ}$ were viable, fertile, and indistinguishable from control littermates.

### Homozygous *Mpdz* mutations cause postnatal lethality

*Mdpz*$^{+/-}$ mice were interbred and we obtained offspring in the expected Mendelian ratio (1:2:1) when animals were genotyped in the embryo–fetal period. Genotyping between postnatal days 1 and 10 (P1–P10) revealed 88 (24%) wild-type, 205 (56%) heterozygous, 73 (20%) knockout mice, indicating a non-Mendelian distribution [$P = 0.038$; chi-square test (Montoliu, 2012)]. Indeed, knockout animals died in the postnatal period and median survival of *Mpdz*$^{-/-}$ mice was 20 days (Fig 1E). No significant sex-specific differences in survival were detectable. Similar data were obtained with CMV-Cre;*Mpdz*$^{Δ/Δ}$ mice. After crossing CMV-Cre;*Mpdz*$^{wt/Δ}$ with *Mpdz*$^{fl/fl}$ mice, offspring CMV-Cre;*Mpdz*$^{Δ/Δ}$ mice were born at expected Mendelian frequency of 25% [$n = 167$; $P = 0.02$; chi-square test (Montoliu, 2012)] and died during the first month of life (median survival 25 days) (Fig 1F).

*Mpdz*$^{-/-}$ mouse pups looked unremarkable at P1 and P2. There were no obvious phenotypic differences compared to littermate controls. At P3, some of the homozygous mice could already be recognized due to a slightly enlarged and dome-shaped skull. Brains isolated from *Mpdz*$^{-/-}$ mice at P4 were enlarged (Fig 1G). From P5 onward, *Mpdz*$^{-/-}$ mice showed growth retardation and increased frequency and severity of neurological symptoms (decreased alertness, lethargy, movement disorders, muscle weakness, apathy). Therefore, all mice were sacrificed upon exhibition of movement disorders or lethargy.

Clinical autopsy and histopathological examination of *Mpdz*$^{-/-}$ mice at P7 revealed macrocephaly, enlarged hemispheres, and massive CSF accumulation in lateral brain ventricles. Hydrocephalus was never detected in any of the wild-type or heterozygous mice. CSF was clear and there were no signs of brain hemorrhage. Compared to wild-type littermates, no obvious alterations in heart, liver, kidney, and gut were detectable. This resembles findings from patients with homozygous *MPDZ* mutations (Al-Dosari *et al*, 2013).

CMV-Cre;*Mpdz*$^{Δ/Δ}$ mice also developed macrocephaly and severe CSF accumulation in lateral brain ventricles, indicating that truncation of Mpdz acts as a null allele (Fig 1H). In older mice (P21) with advanced hydrocephalus, brain hemorrhages were frequently observed.

**Figure 1. Generation of global and conditional *Mpdz* knockout mice.**

A    Schematic representation of the *Mpdz* gene locus and the translated protein. The numbered boxes represent exons 1–47 of the wild-type allele. The relative position of the translational start and stop sites is indicated at exon 2 and exon 47, respectively. A gene-trap cassette (β-geo) was inserted into intron 11–12 leading to a stop signal that truncates the Mpdz protein after the third PDZ domain.

B    Western blotting to detect Mpdz protein in brain lysates of littermate wild-type and global knockout *Mpdz* mice.

C    Schematic drawing of the conditional *Mpdz* gene targeting strategy. LoxP sites flanking exon 4 and a neomycin resistance cassette were inserted by homologous recombination. Transgenic mice were crossed with Flp deleter mice to remove the FRT-flanked neomycin cassette. Cre recombinase removes exons 4 and 5 leading to a frameshift and a nonsense mutation. This truncates the Mpdz protein after the L27 domain.

D    Western blotting to detect Mpdz protein in brain lysates of $Mpdz^{fl/fl}$ and CMV-Cre;$Mpdz^{\Delta/\Delta}$ mice.

E, F    Kaplan–Meier survival analysis of $Mpdz^{-/-}$ vs. $Mpdz^{+/+}$ mice ($n > 41$ mice per genotype) and CMV-Cre;$Mpdz^{\Delta/\Delta}$ vs. $Mpdz^{fl/fl}$ mice ($n > 48$ mice per genotype).

G, H    Representative images of $Mpdz^{-/-}$ and $Mpdz^{+/+}$ mice at postnatal day 27 (P27) and CMV-Cre;$Mpdz^{\Delta/\Delta}$ and $Mpdz^{fl/fl}$ mice at P24. Arrows indicate the enlarged and dome-shaped skull and the enlarged brain hemispheres.

Source data are available online for this figure.

## $Mpdz^{-/-}$ mice develop hydrocephalus after birth

Computed tomography showed massive enlargement of the skull, rupture of sutures, and thinning of skull bones in $Mpdz^{-/-}$ animals at P27 (Fig 2A). Magnetic resonance imaging revealed massive accumulation of CSF in the lateral ventricles with severe thinning of the cerebral cortex and severe compression of the brain stem and cerebellum (Fig 2B). This finding was very similar to the congenital hydrocephalus observed in human patients carrying homozygous *MPDZ* mutations (Al-Dosari et al, 2013).

In humans, severe hydrocephalus was detected in fetuses carrying homozygous *MPDZ* mutations at 30 weeks of gestation (Al-Dosari et al, 2013). Analysis of formalin-fixed brain tissue derived from $Mpdz^{-/-}$ mouse embryos did not show any signs of enlarged brain ventricles at embryonic stage E14.5. Even at birth (P0), no obvious enlargement of the lateral ventricles was visible. At P3, a substantial enlargement of the lateral ventricles occurred, while we did not detect other major changes in brain anatomy. At P7, a severe enlargement of the lateral ventricles, but not of the 4th ventricle, was observed indicating aqueductal stenosis (Fig 3).

## No morphological alterations in the brain vasculature of $Mpdz^{-/-}$ mice

We asked whether the development of hydrocephalus in $Mpdz^{-/-}$ mice might be due to impaired cerebrovascular integrity. H&E staining showed no signs of brain hemorrhage at P3 and P7 (Fig 3). CSF in the enlarged lateral ventricles was clear and did not contain erythrocytes. Therefore, we concluded that hydrocephalus cannot be acquired due to hemorrhage. Staining of endothelial cells with antibodies against CD31 revealed normal vascular patterning and no changes in microvessel density in $Mpdz^{-/-}$ mice compared to littermate controls (Fig EV1A).

Immunohistochemical staining of the vascular tight junction proteins claudin-5 and ZO-1 also showed no differences in cortical vessels of $Mpdz^{-/-}$ mice compared to controls (Fig EV1B).

After P21, we frequently observed brain hemorrhages and blood in CSF of $Mpdz^{-/-}$ and CMV-Cre;$Mpdz^{\Delta/\Delta}$ mice. A substantial increase in hydrostatic pressure is most likely responsible for the rupturing of blood vessels in very advanced stages of hydrocephalus. However, even at P27, electron microscopy analysis of brain sections from $Mpdz^{-/-}$ mice revealed no overt alterations in the composition of blood vessels with numerous tight junctions between endothelial cells (Fig EV1C).

## Endothelial-specific deletion of *Mpdz* does not cause hydrocephalus

To further analyze the potential involvement of the vasculature in hydrocephalus development of *Mpdz*-deficient mice, we ablated *Mpdz* specifically in endothelial cells using a Tie2-Cre driver line (Constien et al, 2001). Mpdz protein expression was strongly reduced in isolated lung endothelial cells derived from Tie2-Cre; $Mpdz^{\Delta EC/\Delta EC}$ mice (Fig EV1D). Tie2-Cre;$Mpdz^{\Delta EC/\Delta EC}$ mice were viable and did not show any signs of hydrocephalus ($n > 200$ transgenic animals). In addition, adult Tie2-Cre;$Mpdz^{\Delta EC/\Delta EC}$ mice (4 months old) appeared healthy and showed normal brain morphology (Fig EV1E).

Taken together, there were no obvious alterations in blood vessel composition and morphology in *Mpdz*-deficient mice and endothelial-specific *Mpdz* ablation did not cause hydrocephalus.

## Normal development of the choroid plexus

The blood–CSF barrier requires very dense tight junctions between epithelial cells at the choroid plexus (Wolburg & Paulus, 2010; Liddelow, 2015), where *Mpdz* is abundantly expressed (Ullmer et al, 1998; Bécamel et al, 2001; Sitek et al, 2003). Choroid plexus epithelial cells transport ions from the plasma ultrafiltrate into the ventricular lumen to generate CSF (Liddelow, 2015). Scanning electron microscopy analysis of the choroid plexus in the lateral ventricles of $Mpdz^{-/-}$ mice at P7 revealed that epithelial cells and microvilli on their apical surface were indistinguishable to wild-type littermates (Fig EV2A). Electron microscopy of ultrathin sections showed normal cell morphology and numerous tight junctions between choroid plexus epithelial cells. As well, the fenestrated endothelial cells of the capillaries in the stroma of the choroid plexus appeared normal at P3 and P7 (Fig EV2B).

Cerebrospinal fluid analysis in mice is extremely challenging due to the very small volume. Also, the amount of CSF derived from $Mpdz^{-/-}$ mice at P3 (mild hydrocephalus) precluded chemical tests. In advanced stages of hydrocephalus (P10–P12), the lateral ventricles were filled with up to 100 μl of CSF. Here, total CSF protein levels were $0.59 \pm 0.18$ g/l ($n = 5$ mice; only such without obvious brain hemorrhage and clear CSF). In very advanced stages of hydrocephalus (P27), CSF protein levels were $1.12 \pm 0.11$ g/l ($n = 5$ mice; only such without obvious brain hemorrhage and clear CSF). While there is no reference range for mice, in rats total protein in CSF is less than 0.35 g/l (Lehtinen et al, 2011). The increased CSF protein concentration may indicate secondary brain damage due to hydrocephalus, but does not allow drawing any conclusions about the pathogenesis of hydrocephalus *per se*.

## Aqueductal stenosis after postnatal day 3

Cerebrospinal fluid produced by choroid plexus in the lateral ventricles passes through the foramina of Monro into the 3rd ventricle, followed by passage through the cerebral aqueduct into the 4th ventricle. To analyze the flow of CSF through the ventricular system in $Mpdz^{-/-}$ mice, Evans blue dye was injected into the right lateral ventricle. When injected at P1, the tracer could be observed in the contralateral ventricle as well as in the 3rd and 4th ventricles and in the spinal subarachnoid space 5 min after injection (Fig 4A, upper row). There were no differences between $Mpdz^{-/-}$ and wild-type littermate controls. This shows again that there is no aqueductal stenosis in newborn $Mpdz^{-/-}$ mice. However, at P3, Evans blue reached the 3rd and the contralateral ventricle, but only very little or no tracer could be detected in the 4th ventricle of $Mpdz^{-/-}$ mice. At P5, this was even more pronounced, indicating a complete blockage of tracer flow through the cerebral aqueduct (Fig 4A). These data verify that the development of a macroscopically visible hydrocephalus from P3 onward is associated with aqueductal stenosis. We thus focused on determining the potential cause for the lack of flow through the cerebral aqueduct.

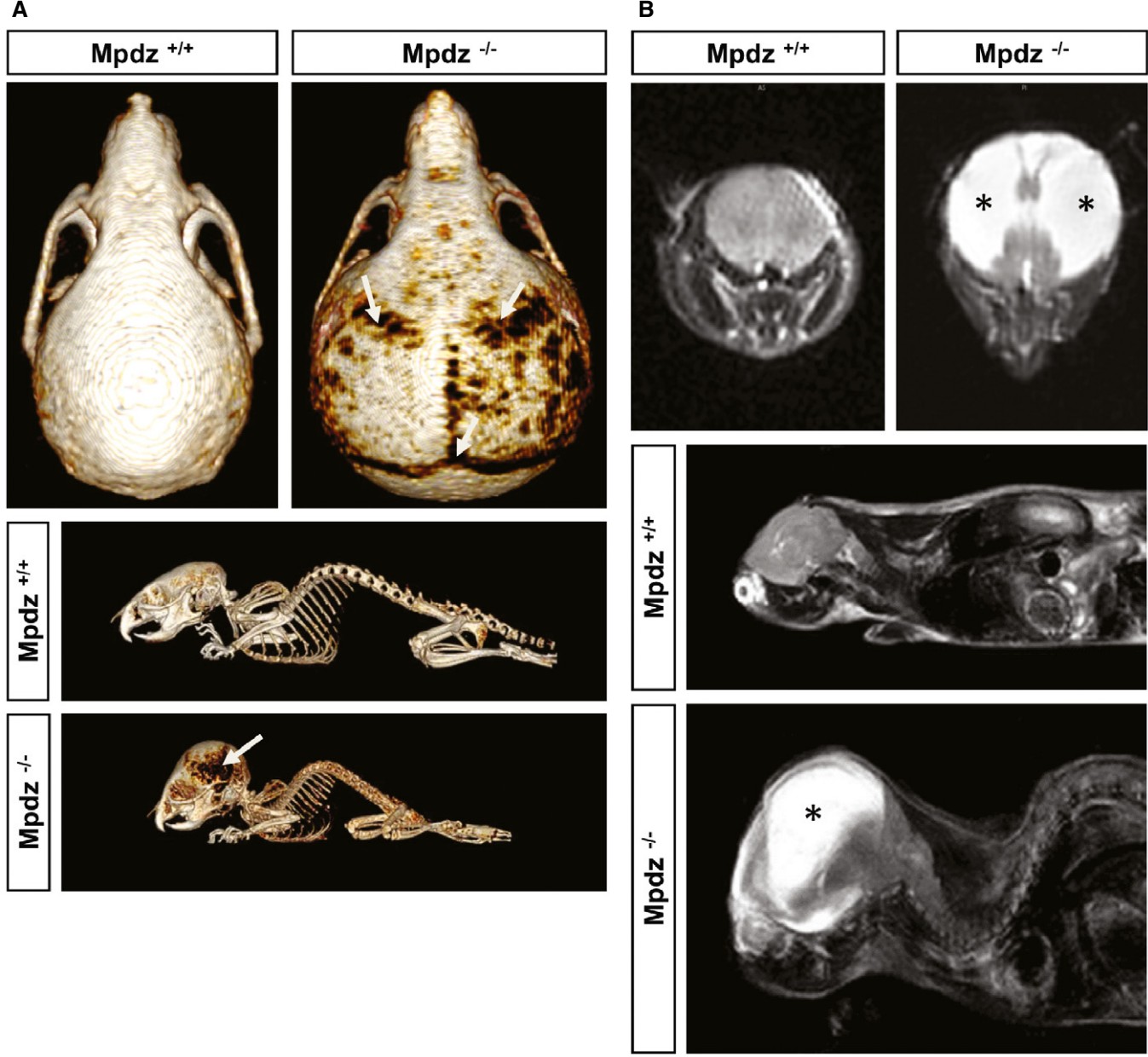

**Figure 2. Mpdz-deficient mice develop hydrocephalus.**

A   At postnatal day 27 (P27), *Mpdz*[-/-] and wild-type littermates were subjected to a computed tomography. Three-dimensional reconstruction shows macrocephaly and thinning of skull bones (arrows).

B   T2-weighted coronal and sagittal magnetic resonance images of the head of *Mpdz*[-/-] vs. *Mpdz*[+/+] mice at P27. CSF in the enlarged lateral ventricles appears hyperintense (asterisks).

### No overt cilia dysfunction in *Mpdz*-deficient brains

In humans, CSF flow is commonly ascribed to a pressure gradient generated by blood pulsations between the sites of production and absorption. However, it is also assumed that CSF movement through the ventricular system is facilitated by motile 9 + 2 cilia at the apical cell membrane of ependymal cells (Lee, 2013; Jiménez *et al*, 2014). In mice, motile cilia generate CSF flow from P8 onward (Tissir *et al*, 2010; Siyahhan *et al*, 2014), and cilia dysfunction

causes postnatal hydrocephalus (Davy & Robinson, 2003; Ibañez-Tallon *et al*, 2004; Banizs *et al*, 2005; Lechtreck *et al*, 2008; Jacquet *et al*, 2009; Wodarczyk *et al*, 2009; Tissir *et al*, 2010; Liu *et al*, 2014; Koschützke *et al*, 2015; Rachel *et al*, 2015).

Scanning electron microscopy revealed normal numbers of cilia on the ependymal cell surface within the lateral ventricles of *Mpdz*[-/-] mice at P7 (Fig 4B). In accordance with normal cilia morphology, we could not detect typical signs of a ciliopathy in *Mpdz*[-/-] mice, such as cysts in liver or kidney (Fig EV2C).

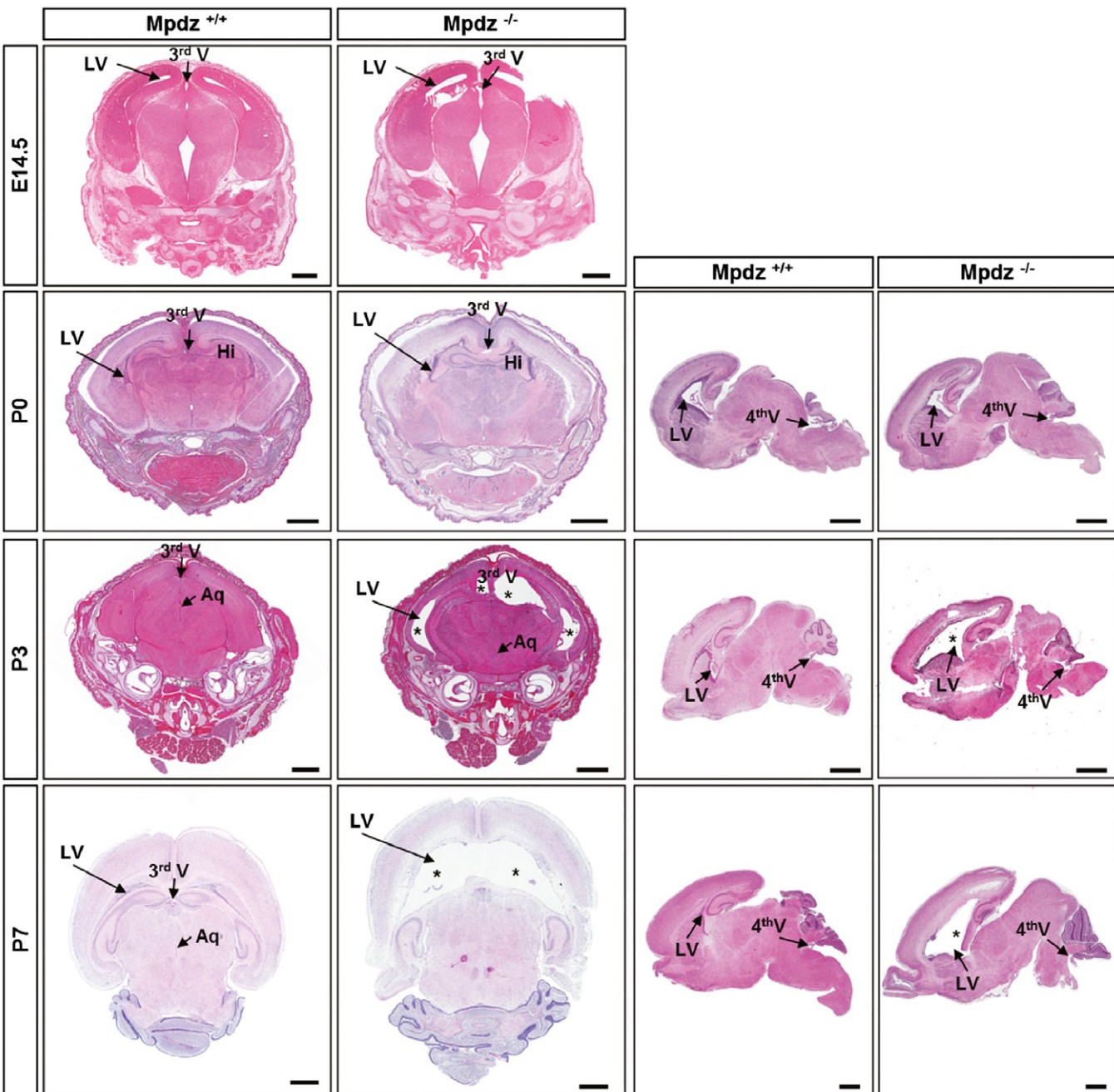

**Figure 3.  *Mpdz*⁻ᐟ⁻ mice develop postnatal hydrocephalus.**
H&E staining of brain sections from *Mpdz*⁻ᐟ⁻ and littermate *Mpdz*⁺ᐟ⁺ mice at different developmental stages. At embryonic stage E14.5 and at birth (P0), coronal sections showed no overt alterations in *Mpdz*⁻ᐟ⁻ mouse brains. At P3 and P7, enlarged lateral ventricles (*) were detected in *Mpdz*⁻ᐟ⁻ mice. Horizontal sections demonstrate ventriculomegaly in *Mpdz*⁻ᐟ⁻ mice at P3 and P7. Aq, cerebral aqueduct; Hi, hippocampus; LV, lateral ventricle; V, ventricle (3ʳᵈ, 4ᵗʰ). Scale bars: E14.5, 500 μm; P0, P3, and P7, 1 mm.

Serum analyses also showed no pathological alterations of creatinine, urea, uric acid, and albumin serum levels and no significant differences between *Mpdz*⁻ᐟ⁻ and control mice at P14. This indicates that renal function was not impaired in *Mpdz*⁻ᐟ⁻ mice. Taken together, the normal cilia morphology and the fact that motile cilia generate significant CSF flow in mice only from P8 and onward (Tissir *et al*, 2010) dismisses the idea of cilia

dysfunction as the primary cause for hydrocephalus development in *Mpdz*⁻ᐟ⁻ mice.

Next we performed immunostaining to detect apoptotic cells by the presence of active caspase-3 in the choroid plexus, the ependyma, and the subventricular zone. This revealed that at P0, there were very few apoptotic cells detectable. However, at P3 there were substantially more cells positive for cleaved caspase-3 in the choroid

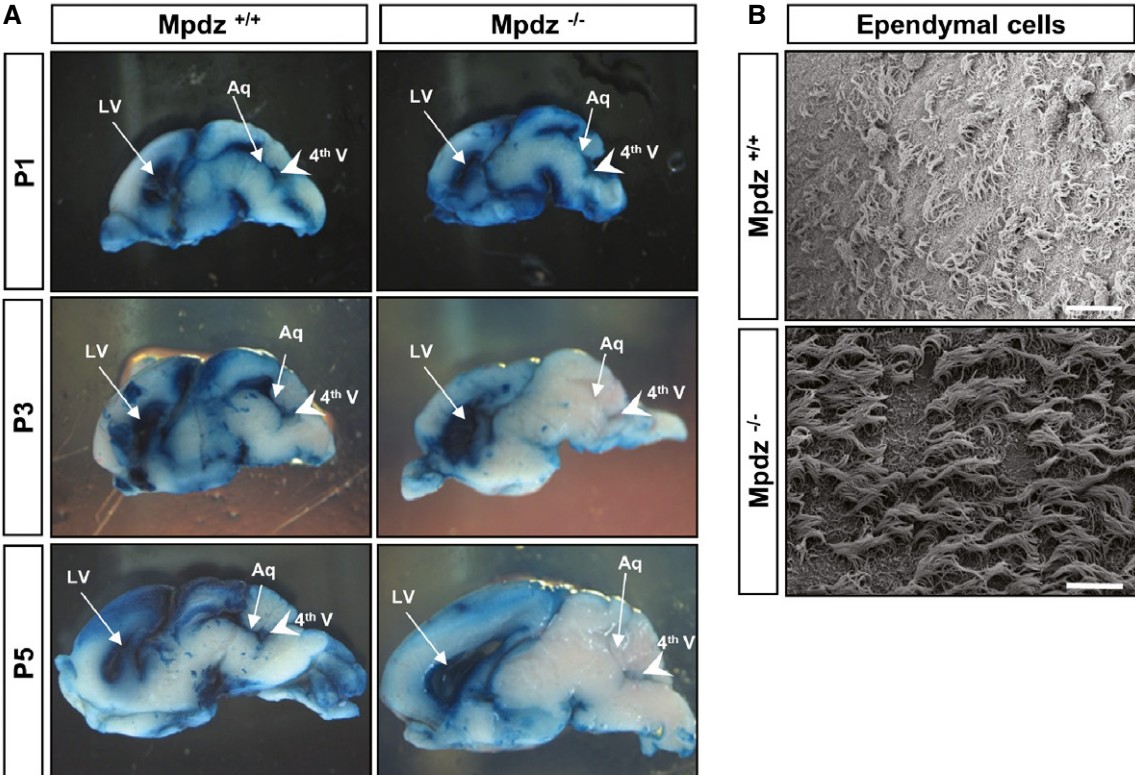

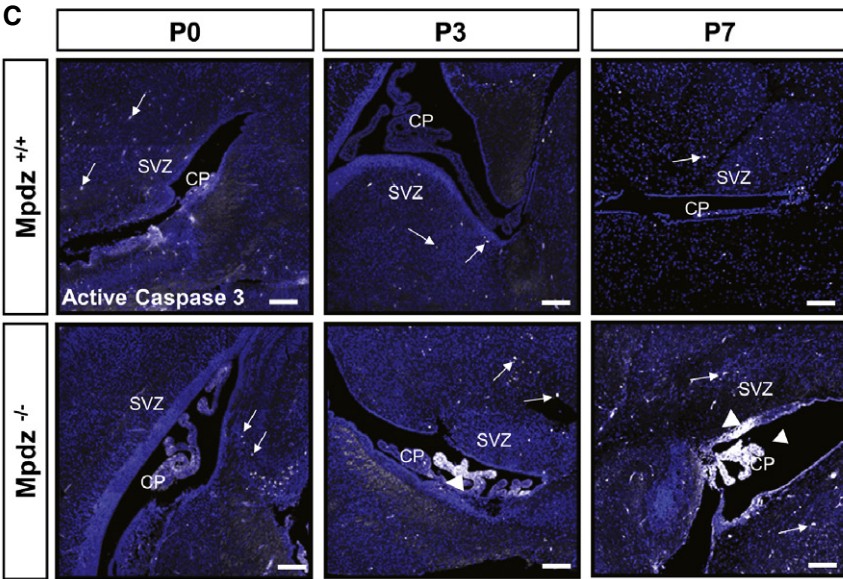

**Figure 4.  Impaired flow of CSF through the cerebral aqueduct after postnatal day 3.**

A   Flow of Evans blue dye injected into a lateral ventricle trough the ventricular system of *Mpdz*−/− and *Mpdz*+/+ littermates was analyzed at postnatal day 1 (P1), P3, and P5. Sagittal sections show the lateral (LV) and fourth ventricles (4th V, arrowhead). In mice at P1, flow of Evans blue from one lateral ventricle through the third into the contralateral and the fourth ventricles was unremarkable in *Mpdz*−/− and *Mpdz*+/+ mice. At P3 and P5, no flow of Evans blue through the cerebral aqueduct (Aq) into the fourth ventricle could be detected in *Mpdz*−/− mice.

B   Scanning electron micrograph of ependymal cells lining the roof of lateral ventricles at P7. No morphological alterations in motile cilia of *Mpdz*−/− compared to *Mpdz*+/+ mice. Scale bar, 20 μm.

C   Antibody staining against cleaved caspase-3 (white color, arrows and arrowheads) to detect apoptotic cells. CP, choroid plexus; SVZ, subventricular zone. Scale bar, 100 μm.

plexus and the ependyma of $Mpdz^{-/-}$ mice compared to controls. At P7, we detected strong caspase-3 activity within the ependymal cell layer and throughout the choroid plexus (Fig 4C).

## Loss of *Mpdz* leads to defects in the ependyma

The ventricular system is lined by the ependyma, a single layer of simple cuboidal to columnar epithelium with microvilli and motile cilia on the apical surface. *Mpdz* expression is very pronounced in this cell layer (Bécamel *et al*, 2001; Sitek *et al*, 2003), which separates CSF from brain tissue. Immunostaining indicated higher rates of cell death in the ependyma of $Mpdz^{-/-}$ compared to wild-type littermate controls (Fig 4C). H&E staining revealed that at P0, there were some sporadic defects within the ependymal layer of the lateral ventricles of $Mpdz^{-/-}$ mice. Most notably, we found severe defects in the ependymal lining of the cerebral aqueduct. Here single cells or a whole stretch of ependymal cells was missing. At P3 and P7, this was even more severe. In parts of the cerebral aqueduct, ependymal denudation had occurred (Fig 5). This indicates that loss of tight junction-associated protein Mpdz disturbs the integrity of the ependymal layer leading to partial ependymal denudation.

## Inactivation of *Mpdz* with Nestin-Cre leads to hydrocephalus formation

To further test this hypothesis, we inactivated the *Mpdz* gene in radial glia cells, the precursors for ependymal cells, and neuronal precursors using the Nestin-Cre driver line (Tronche *et al*, 1999). Indeed, Nestin-Cre$^{+/+}$;$Mpdz^{\Delta/\Delta}$ mice developed hydrocephalus in the early postnatal period. Very similar to the global $Mpdz^{-/-}$ mice, there was severe enlargement of the lateral ventricles (Fig EV3).

## Loss of *Mpdz* leads to impaired epithelial barrier function

As the loss of *Mpdz* induced major changes in the ependyma, we were wondering whether *Mpdz* might have a function in epithelial barrier control. It is important to note that presence and normal morphological appearance of tight junctions do not exclude a barrier defect. This had been shown, for example, in occludin (*Ocln*)- or claudin-5 (*Cldn5*)-deficient mice, in which tight junctions are still formed but are not fully functional (Saitou *et al*, 2000; Nitta *et al*, 2003). Therefore, we analyzed cellular permeability after silencing of *Mpdz* expression. Due to a lack of suitable ependymal cell lines and the difficulty to culture suitable amounts of primary ependymal cells from mice, we employed human MCF7 cells as surrogate model to study cellular barrier functions. *MPDZ* expression was silenced by transfection with siRNAs or by transduction with lentiviral particles expressing shRNAs by $80 \pm 7\%$. Cellular permeability was determined by measuring transepithelial electrical resistance (TER) and corresponding capacitance ($C_{cl}$). Once cells had reached full confluence, as indicated by low $C_{cl}$, we observed severely reduced TER in MCF7 cells, indicating impaired barrier properties (Fig 6A–C). These findings could be verified with human choroid plexus epithelial papilloma (HIBCPP) cells. HIBCPP cells form tight junctions, develop a high electrical resistance and minimal levels of macromolecular flux when grown on transwell filters, and thereby represent an excellent model system for the blood–

cerebrospinal fluid barrier (Schwerk *et al*, 2012). Silencing of *MPDZ* expression led to lower TER indicating impaired barrier function of HIBCPP cells (Fig 6D).

Mpdz binds to the RhoA-specific guanine exchange factor Syx (PLEKHG) (Estévez *et al*, 2008), and recruits it to the Pals1 polarity complex. Thereby, Mpdz is involved in controlling the activity of RhoA. This small G protein can remodel the cytoskeleton and cell–cell junctions leading to increased endothelial permeability (Wu *et al*, 2011; Ngok *et al*, 2012). In oligodendrocytes, NG2 stimulates RhoA activity at the cell periphery via Mpdz (MUPP1) and Syx1 (Biname *et al*, 2013). Notably, increased Rho kinase activity after deletion of myosin IXa can lead to hydrocephalus in mice (Abouhamed *et al*, 2009). To test whether Mpdz also controls RhoA activity in cells of the CNS, we analyzed its activity in primary astrocytes isolated from neonatal mouse brains. Astrocyte lysates derived from $Mpdz^{-/-}$ mice had significantly higher RhoA activity levels compared to those derived from wild-type littermate controls (Fig 6E).

Taken together, the data showed that *Mpdz* is critical to maintain an intact ependymal cell layer and epithelial barrier integrity.

## Disturbed Pals1 expression in $Mpdz^{-/-}$ mice

Mpdz is associated with components of the planar cell polarity complex via binding to Pals1 (Wu *et al*, 2011), and disturbance of planar cell polarity can lead to hydrocephalus in mouse models (Tissir *et al*, 2010; Ohata *et al*, 2014). We detected that the onset of Pals1 expression in the ependymal cell layer is delayed in neonatal $Mpdz^{-/-}$ mice. Whereas wild-type littermate controls showed robust Pals1 expression in ependymal cells of the lateral ventricles already at P0, this was almost absent in $Mpdz^{-/-}$ mice. Only from P7 on there was Pals1 expression detectable in ependymal cells but still not continuous through the ependymal layer (Fig 7A). However, the expression levels and dynamic expression pattern of the planar cell polarity protein Crb3 were unremarkable in $Mpdz^{-/-}$ mice (Fig 7B). Also, the expression pattern of adherens junction protein E-cadherin was not altered in $Mpdz^{-/-}$ mice (Fig 7C).

## Hydrocephalus development is preceded by astrogliosis

Defects of the ependyma lead to reactive astrogliosis, which can obstruct the cerebral aqueduct (Wagner *et al*, 2003). Therefore, we tested whether ependymal defects in $Mpdz^{-/-}$ mice are accompanied by astrogliosis. This repair mechanism is characterized by enhanced glial fibrillary acidic protein (GFAP) expression, cellular hypertrophy, and astrocyte proliferation (Fawcett & Asher, 1999). At P0, when no hydrocephalus was present, GFAP expression was prominent in the ependyma of the lateral ventricles and in the subependymal zone of $Mpdz^{-/-}$ mice. Strong GFAP expression was also observed in the hippocampal region of $Mpdz^{-/-}$ but not in control mice. Most importantly, we observed astrogliosis within the cerebral aqueduct of $Mpdz^{-/-}$ mice (Figs 8 and EV4). This clearly indicates that these cells are subjected to stress or damage prior to the development of a hydrocephalus.

At P3, we detected much stronger GFAP expression in the ependyma and the subependymal zone in $Mpdz^{-/-}$ mice compared to control, which was even more pronounced at P7. In particular, the cerebral aqueduct was affected. In addition, strong astrogliosis in $Mpdz^{-/-}$ mice was observed in the hippocampal region of *Mpdz*-

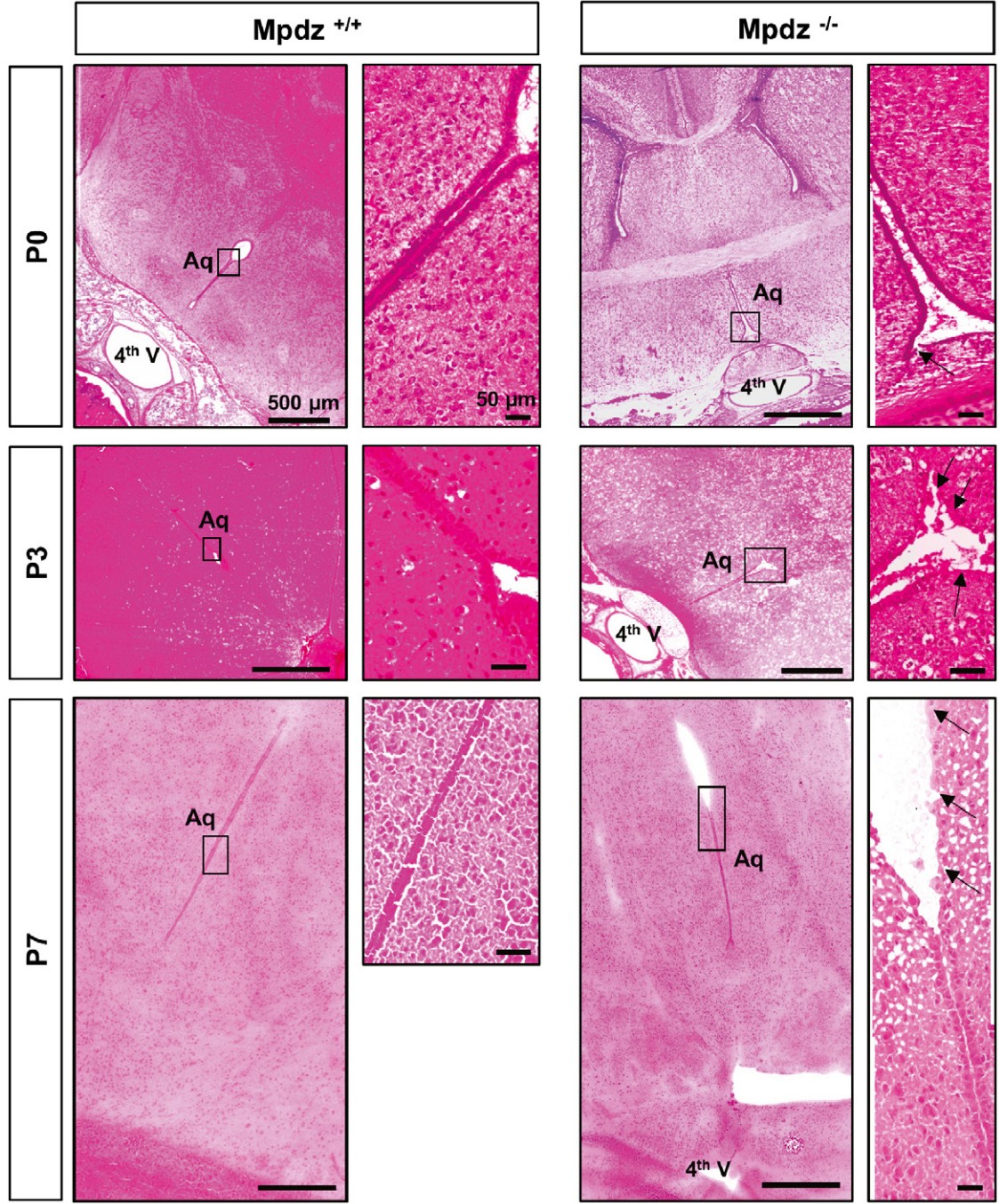

**Figure 5. Ependymal defects in the cerebral aqueduct of *Mpdz*$^{-/-}$ mice.**
H&E staining of horizontal brain sections from of *Mpdz*$^{-/-}$ and *Mpdz*$^{+/+}$ littermates at postnatal days 0, 3, and 7 (P0, P3, and P7). A continuous single layer of ependymal cells lines the aqueduct in controls. Disruption of the ependymal cell layer was frequently observed in *Mpdz*-deficient mice. The images at the right show areas of interest (boxed) in higher magnification. Arrows indicate regions where the ependymal layer is disturbed. Please note that images of *Mpdz*$^{+/+}$ brains (P0, P3) are close-ups of the images shown in Fig 3. Scale bars are indicated in images.

deficient mice at P7 (Figs 8 and EV4). Gliosis within the cerebral aqueduct narrows the lumen and impairs CSF flow (Cinalli *et al*, 2011).

In summary, the data show that astrogliosis—secondary to ependymal cell dysfunction—obstructs CSF flow through the cerebral aqueduct leading to extreme dilation of the supratentorial ventricular system in *Mpdz*$^{-/-}$ mice.

## Discussion

*MPDZ* is the only human gene locus known to be associated with autosomal recessive non-syndromic congenital hydrocephalus (Al-Dosari *et al*, 2013). We generated ubiquitous and cell type-specific mouse models verifying that homozygous truncating *Mpdz*

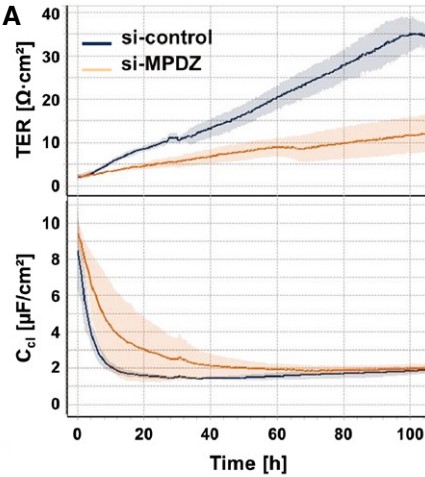

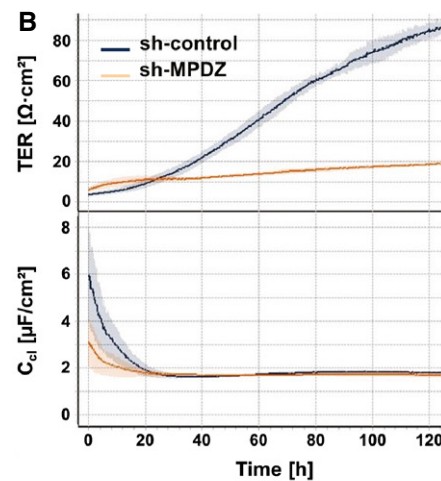

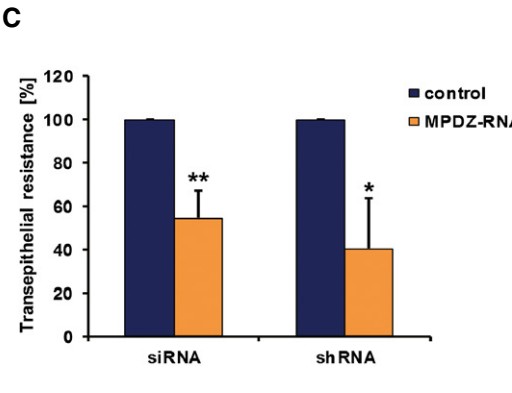

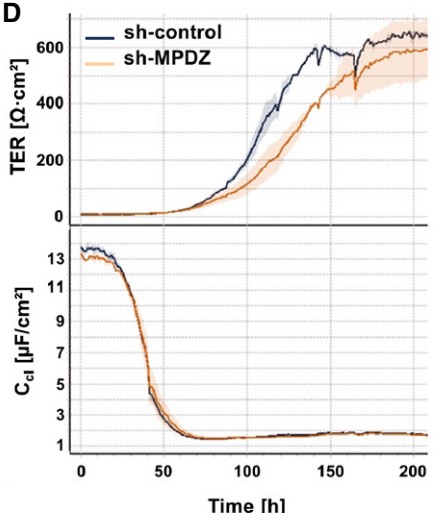

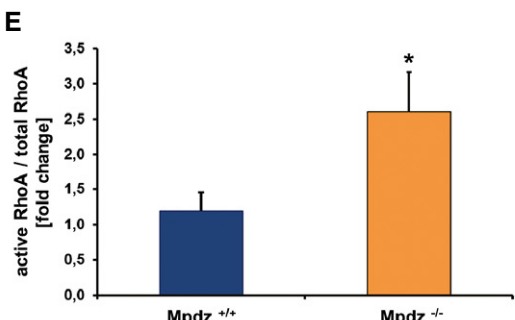

**Figure 6. Loss of MPDZ increases epithelial cell permeability.**

A, B    Representative graph of transepithelial electrical resistance (TER; upper graph) and corresponding capacitance ($C_{cl}$; lower graph) of MCF-7 monolayers for 5 days after silencing of *MPDZ* expression. The reduction of *MPDZ* expression led to a significant decrease of TER indicating increased permeability. The low corresponding capacitance ($C_{cl}$) values indicate formation of a dense cell monolayer. $n$ = 3 technical replicates.

C    Mean TER values of five (siRNA) or three (shRNA) independent experiments with each three replicates at time point 72 h. **$P$ = 0.008; *$P$ = 0.0113 by two-sided, unpaired Student's $t$-test.

D    Representative graph of TER and corresponding capacitance ($C_{cl}$) of HIPCC cell monolayers for 5 days after silencing of *MPDZ* expression. $n$ = 3 replicates.

E    Levels of active RhoA (RhoA-GTP) and total RhoA levels in isolated astrocytes derived from brains of neonatal *Mpdz$^{-/-}$* and *Mpdz$^{+/+}$* mice were detected by G-LISA. Graph shows ratio of active RhoA vs. total RhoA. $n$ = 3; $P$ = 0.0171 by two-sided, unpaired Student's $t$-test.

Data information: All data are presented as mean ± SD.
Source data are available online for this figure.

**Figure 7. Diminished expression of Pals1 in ependymal cells of *Mpdz*$^{-/-}$ mice.**

Immunostaining for the planar cell polarity proteins Pals1 and Crb3 (white color) and the adherens junction protein E-cadherin (brown color) in brains of *Mpdz*-deficient mice at postnatal days 0, 3, and 7 (P0, P3, and P7).

A   Expression of Pals1 (white color, arrows) is absent (*) or diminished in *Mpdz*$^{-/-}$ mice. Scale bars: 10 μm for P0, 20 μm for P3, 10 μm for P7.

B   Expression of Crb3 is not altered in *Mpdz*$^{-/-}$ compared to control mice. Scale bars: 10 μm for P0, 20 μm for P3, 10 μm for P7.

C   The dynamic expression pattern of E-cadherin is not altered in *Mpdz*$^{-/-}$ compared to control mice. Scale bars: 100 μm and for zoom-ins 10 μm.

Data information: CP, choroid plexus; EP, ependymal cells; LV, lateral ventricle.

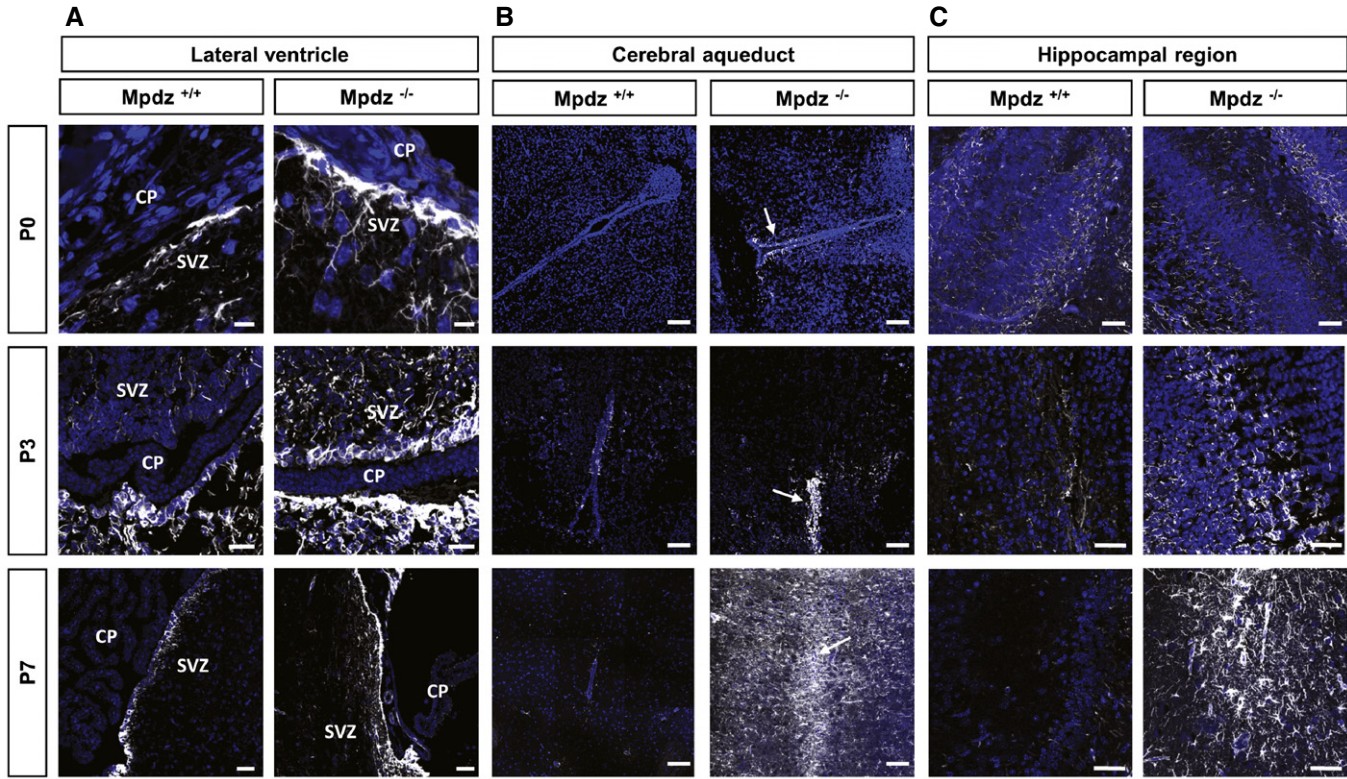

**Figure 8. Hydrocephalus in *Mpdz*-deficient mice is preceded by reactive gliosis.**

Immunostaining for glial fibrillary acidic protein (GFAP; white color, arrows).

A   Progressive astrogliosis in the subventricular zone (SVZ) of *Mpdz*$^{-/-}$ mice. CP, choroid plexus. Scale bars: 10 μm for P0, 40 μm for P3, 50 μm for P7.

B   Progressive astrogliosis (indicated by arrow) around the cerebral aqueduct of *Mpdz*$^{-/-}$ mice. Scale bars: 100 μm.

C   Progressive astrogliosis in the hippocampal region of *Mpdz*$^{-/-}$ mice. Scale bars: 100 μm for P0, 50 μm for P3 and P7.

mutations cause hydrocephalus. *Mpdz*-deficient mice developed supratentorial hydrocephalus with ventriculomegaly leading to macrocephaly and compression of the brain cortex and brain stem. In contrast to human patients (Al-Dosari *et al*, 2013), this occurred only after birth in *Mpdz*-deficient mice. Interestingly, *L1cam*-deficient mice also develop postnatal hydrocephalus, while in humans carrying *LCAM1* mutations hydrocephalus often develops already before birth (Rolf *et al*, 2001).

MPDZ localizes to tight junctions and directly interacts with several other tight junction proteins (e.g., claudin-1, JAM-A, ZO-3) (Adachi *et al*, 2009). There is some evidence that alterations in cell–cell junctions within neuroepithelial cells might be causative for congenital hydrocephalus (Jiménez *et al*, 2014; Guerra *et al*, 2015). Our study demonstrates that *Mpdz* is not essential for tight junction formation *in vivo*. Consistently, there were also no signs of cell polarity defects in *Mpdz*$^{-/-}$ mice. This may be due to compensation by increased expression of the related INADL protein upon loss of MPDZ (Adachi *et al*, 2009; Assémat *et al*, 2013). We observed a significant increase in *INADL* mRNA expression upon silencing of *MPDZ* in endothelial cells (HUVEC and isolated brain endothelial cells from *Mpdz*$^{-/-}$ mice) (Appendix Fig S1). Simultaneous targeting of both genes will unravel this issue in the future. However, it is important to note that although tight junctions were formed upon inhibition of *Mpdz* expression, epithelial barrier integrity was

disturbed. This is reminiscent of other knockout models of tight junction molecules (e.g., *Jam3*-, occludin (*Ocln*)-, or claudin-5 (*Cldn5*)-deficient mice), in which tight junctions are present but are not fully functional (Saitou *et al*, 2000; Nitta *et al*, 2003; Wyss *et al*, 2012). As such, Mpdz appears to be a critical protein for fine-tuning the tightness of cell–cell junctions.

Similar as in *Jam3*-deficient mice (Wyss *et al*, 2012), the endothelium was not responsible for hydrocephalus development in *Mpdz*$^{-/-}$ mice. Based on the presented data, we propose that loss of *Mpdz* disturbs integrity of the ependymal cell layer. Loss of either cell–cell or cell–matrix interactions would trigger apoptosis, as observed by increased caspase-3 activity in the ependyma of *Mpdz*$^{-/-}$ mice. *Mpdz* inactivation using the Nestin-Cre line further indicated that defects in the ependyma are the primary cause of hydrocephalus. *Nestin* is expressed in glial and neuronal precursors. Deleting *Mpdz* specifically in cilia-containing ependymal cells by using the FoxJ1-Cre line (Meletis *et al*, 2008) will finally resolve this issue.

Mechanistically, we could show that expression of the planar cell polarity protein Pals1 is strongly reduced in ependymal cells of newborn *Mpdz*$^{-/-}$ mice. Pals1 directly interacts with Mpdz and also stabilizes this protein (Assémat *et al*, 2008, 2013). Therefore, one could assume that Mpdz helps to stabilize Pals1 expression at the membrane of ependymal cells. Defects in proteins of the planar cell polarity complex can lead to hydrocephalus formation in mice

(Tissir *et al*, 2010; Ohata *et al*, 2014). The expression of another cell polarity protein Crb3 was not affected in the ependyma of *Mpdz*$^{-/-}$ mice, indicating not a general impairment of planar cell polarity. Pals1 is linked to the RhoA-specific guanine exchange factor Syx via the adaptor protein Mpdz (Estévez *et al*, 2008). As such, Mpdz is involved in the control of RhoA activity (Wu *et al*, 2011; Ngok *et al*, 2012; Biname *et al*, 2013). RhoA-dependent kinases play a major role in controlling the stability of cell–cell junctions. Increased RhoA activity after deletion of myosin IXa (*Myo9a*), a motor molecule with a Rho GTPase-activating (GAP) domain, leads to hydrocephalus formation in mice (Abouhamed *et al*, 2009). The *Myo9a* knockout model shows many similarities with *Mpdz*$^{-/-}$ mice with distortion of the ependymal cell layer, stenosis of the aqueduct, and dilation of the lateral and third ventricles. Similar as observed after silencing of *Mpdz* expression, there was increased RhoA activity in cultured epithelial cells after silencing of *Myo9a* (Abouhamed *et al*, 2009).

Disturbance of ependymal integrity induces hydrocephalus in other mouse models, for example, mice deficient for *Myo9a, Pkcl, Numbl, Ophn1, Myh4, Dlg5,* or *Mdnah5* (Ibañez-Tallon *et al*, 2004; Imai *et al*, 2006; Kuo *et al*, 2006; Khelfaoui *et al*, 2007; Ma *et al*, 2007; Nechiporuk *et al*, 2007; Abouhamed *et al*, 2009). Ependymal damage most likely impairs the brain–CSF barrier and disturbs CSF homeostasis. We suggest that this subsequently results in the initiation of repair processes, in particular reactive astrogliosis. The data indicate that at birth, when no signs of hydrocephalus are yet visible, ependymal defects and reactive gliosis are present in the ependyma and subventricular zone of *Mpdz*$^{-/-}$ mice. Astrogliosis is, in principle, beneficial; however, in the narrow aqueduct, astrogliosis can rapidly obstruct CSF flow into the 4$^{th}$ ventricle. As such, this work showed that slight impairment of ependymal integrity triggers astrogliosis in the subependymal zone, resulting in aqueductal stenosis and ventriculomegaly in *Mpdz*$^{-/-}$ mice.

# Materials and Methods

### Animal experiments

Mice were kept under pathogen-free barrier conditions, and animal procedures were performed in accordance with the institutional and national regulations and approved by the local committees for animal experimentation (Heidelberg University and DKFZ) and the local government (Regierungspräsidium Karlsruhe, Germany). Murine embryonic stem cells containing a gene trap in intron 11–12 of the *Mpdz* gene were obtained from the Mutant Mouse Resource & Research Center (MMRRC) and injected into blastocysts. Anesthesia was administered by intraperitoneal injection of 100 mg/kg ketamine (Ketavet, Pfizer) and 20 mg/kg xylazine (Rompun 2%, Bayer). Chimeric mice were crossed to C57BL/6 mice. Heterozygous *Mpdz*$^{\text{Gt(XG734)Byg(+/-)1AFis}}$ mice (*Mpdz*$^{+/-}$ mice) were further backcrossed to C57BL/6 mice for at least eight generations.

Floxed *Mpdz* mice were generated by homologous recombination in C57Bl/6 × 129/SvEv hybrid ES cells (inGenious Targeting Laboratory, Ronkonkoma, NY, USA) and crossed with mice expressing Cre recombinase under control of the CMV promoter for ubiquitous gene ablation (Schwenk *et al*, 1995). Tie2-Cre mice allowed gene deletion primarily in endothelial cells and a subset of hematopoietic cells (Constien *et al*, 2001) and Nestin-Cre for deletion in radial glia (Tronche *et al*, 1999). All Cre lines were backcrossed into C57Bl/6 for more than 10 generations.

### Magnetic resonance imaging and volumetric computed tomography

Mice were anaesthetized using a mixture of isofluran (1.5%) and oxygen (0.5 l/min). Magnetic resonance (MR) images were acquired on a 1.5-T clinical MR scanner (Symphony, Siemens, Germany) using a homebuilt coil for radiofrequency excitation and detection, designed as a cylindrical volume resonator (*n* = 3 mice per genotype). Morphological T2-weighted images were acquired using a turbo spin-echo sequence (orientation axial, TR 3,240 ms, TE 81 ms, matrix 152 × 256, resolution 0.35 × 0.35 × 1.5 mm$^3$, 3 averages, 15 images, scan time 3:40 min).

Volumetric computed tomography (VCT) imaging was obtained using the following parameters: tube voltage 80 kV, tube current 50 mA, scan time 51 s, rotation speed 10 s, frames per second 120, and slice thickness 0.2 mm. Image reconstructions were performed using a modified FDK (Feldkamp Davis Kress) cone beam reconstruction algorithm (kernel H80a; Afra, Erlangen, Germany). Unenhanced VCT images and MRI-acquired T2-weighted images were analyzed using Osirix Imaging Software.

### Antibodies

For immunofluorescence analysis, the following primary antibodies were used: rabbit anti-ZO-1 (1:100 dilution, Thermo Fisher, #61-7300), rabbit anti-claudin-5 (1:400, Abcam, #ab53765), rabbit anti-GFAP (1:500, DAKO, #Z0334), rat anti-CD31 (clone Mec13.3, BD Biosciences), Pals1 (1:100, Merck, #7-708), rat anti-Crb3 (14F9, Abcam, ab180835), mouse anti-E-cadherin (1:200, BD Bioscience, #610181), and rabbit anti-cleaved caspase-3 (Asp175, 1:100 Cell Signaling, #9661). Secondary antibodies were 3 mg/ml Alexa 546-conjugated goat anti-rabbit IgG and 3 mg/ml Alexa 488-conjugated goat anti-rat IgG (all from Invitrogen). For Western blotting, rabbit anti-MPDZ (1:500, Thermo Scientific, 42-2700) and mouse anti-beta actin (1:2,500, Sigma-Aldrich #A5441) were used.

### Immunofluorescence and histology

Freshly dissected brains were embedded in Tissue-Tek (Sakura, Netherlands), frozen, and stored at −80°C. Sections (7 μm) were fixed in methanol for 20 min at −20°C. For GFAP staining, mice were perfused with 4% paraformaldehyde in PBS through the left ventricle of the heart and tissue was embedded in paraffin. Paraffin was removed from fresh sections (4 μm), and antigen retrieval was carried out in citrate buffer (pH 6.0). Blocking solution was 10% goat serum. Primary and secondary antibodies were diluted in blocking solution. Primary antibodies were incubated overnight at 4°C and secondary antibodies for 1 h at room temperature. Sections were washed with TBS-T between incubation steps and finally mounted with Fluoromount (Dako). Confocal images were obtained using a LSM 700 microscope (Carl Zeiss) and analyzed using the Fiji software.

For histology, paraffin-embedded tissue sections were stained with hematoxylin and eosin. Images were taken with a CellObserver microscope (Carl Zeiss) and analyzed using the ZEN Blue software (Carl Zeiss).

## Electron microscopy

Newborn *Mpdz*-deficient and wild-type littermates ($n = 3$ for each age and genotype) were euthanized and perfused with PBS and subsequently with 4% glutaraldehyde and 4% formaldehyde in 0.1 M cacodylate buffer through the left ventricle of the heart. Dissected brains were fixed in 2.0% glutaraldehyde in 0.05 M cacodylate buffer for 2 h and stained with 1% $OsO_4$ in cacodylate buffer for another 2 h followed by contrasting in 0.5% uranyl-acetate. After dehydration in an ascending series of ethanol and propylene oxide, the samples were flat-embedded in Epon (Serva, Germany). Using an ultramicrotome (Ultracut, Leica, Bensheim, Germany), 0.5-μm- and 50-nm-thin sections were cut. Ultrathin sections were stained with 2% uranyl-acetate for 15 min and contrasted in lead citrate for 5 min, mounted on copper grids, and finally analyzed with a Zeiss-EM910 electron microscope.

For scanning electron microscopy, glutaraldehyde-fixed brains were dehydrated, transferred into hexamethyldisilazane, and slowly air-dried. Samples were cut and coated with gold using the agar sputter coater (Agar Scientific, England). Images were taken using the Hitachi S4500 and analyzed with the digital image processing 2.6 software (Point Electronic, Halle).

## Ventricular dye injections

For intracranial dye injection, mice ($n = 5$ per genotype) were anesthetized by intraperitoneal injection of ketamine and xylazine according to approved experimental protocols. Evans blue dye (5 μl, 1% in PBS) was injected slowly into the lateral ventricle using a 0.1-ml syringe. The syringe was left in the injection site to prevent reflux of fluid. Mice were well euthanized 5 min after the injection and the heads were immediately fixed in 4% paraformaldehyde overnight. Brains were dissected, and photographed with an SMZ800 stereo microscope (Nikon).

## Analysis of serum creatinine, urea, and albumin

Plasma and CSF metabolites were measured in the central laboratory of the University Hospital Heidelberg by the same procedures used and validated for routine diagnostic analysis ($n = 5$ mice). Creatinine, urea, and uric acid were measured enzymatically, and total protein in CSF was measured by precipitation with pyrogallol red on an ADVIA 2400 clinical chemistry XPT analyzer (Siemens Healthcare Diagnostics, Eschborn, Germany).

## Cell culture

MCF7 cells were cultured in DMEM containing 10% fetal calf serum (FCS), 100 units/ml penicillin, and 100 μg/ml streptomycin. Human choroid plexus epithelial papilloma HIBCPP cells were cultured in DMEM/F12 supplemented with 10% FCS, 5 μg/ml insulin, 100 U/ml penicillin, and 100 μg/ml streptomycin. For experiments, cells grown in the standard and the inverted cell culture insert system were used as previously described (Schwerk *et al*, 2012). Three different lentiviral shRNA vectors for silencing MPDZ were obtained from Biocat (V2LHS_3656, 16945 16946). Transduced MCF7 cells were selected with 10 μM blasticidin-S. Transient transfections were performed using Oligofectamine (Life Technologies).

**The paper explained**

**Problem**

Congenital hydrocephalus, an abnormal accumulation of CSF in brain cavities, is diagnosed in ~1 of 2,000 newborns. A genetic etiology is assumed for a large proportion of patients. Mutations that cause non-syndromic congenital hydrocephalus in humans have been detected in only two genes: *L1CAM* and *MPDZ*. The function of *MPDZ* is not fully understood.

**Results**

Mouse models to inactivate the *Mpdz* gene were generated, and these mimic the human pathology. *Mpdz* is needed to maintain the integrity of the ependymal cell layer in the brain that forms the cerebrospinal fluid–brain barrier. Loss of *Mpdz* leads to detachment of ependymal cells resulting in astrogliosis, a repair process that leads to obstruction of CSF transport through the cerebral aqueduct.

**Impact**

The work provides a fully penetrant and clinical relevant model to study the pathogenesis of non-syndromic congenital hydrocephalus.

Subsequently, MCF7 cells were seeded onto transwell inserts (0.4 μm pore diameter, Greiner BioOne) and used for transepithelial resistance measurements (cellZscope, nanoAnalytics). Medium was exchanged every second day. Astrocytes from neonatal mice (P2) were isolated and cultured as described (Reischl *et al*, 2014).

Standardized multiplex cell contamination and cell line authentication testing (Multiplexion, Germany) were conducted on a regular basis.

## Quantitative real-time PCR

Total RNA was isolated with the innuPREP RNA Mini Kit (Jena Analytics) and transcribed into cDNA (High Capacity cDNA Reverse Transcription Kit; Applied Biosystems). cDNA was mixed with POWER SYBR Green Master Mix and qPCR performed using an ABI StepOnePlus cycler (Applied Biosystems). The housekeeping genes *GAPDH, RPL32,* and *OAZ1* were used for normalization. Primer sequences are provided upon request. All experiments included two technical and three biological replicates.

## RhoA activity analysis

RhoA activation (ratio RhoA-GTP/total RhoA) was determined using the RhoA G-LISA Activation Assay (Cytoskeleton Inc., BK124).

## Statistical analyses

Results are expressed as means plus/minus standard deviation. Comparisons between groups were made by a two-sided, unpaired *t*-test. Comparison between multiple groups was made by ANOVA. Alterations in Mendelian distribution were calculated with the software tool Mendel.xls based on a chi-square test (Montoliu, 2012). *P*-values < 0.05 were considered as significant.

## Study approval

All animal work was approved by the local committees for animal experimentation (Heidelberg University and DKFZ) and the local

government (Regierungspräsidium Karlsruhe, Germany). This work is not considered "Human Subjects Research".

**Expanded View** for this article is available online.

## Acknowledgements

We thank Katharina Neumeier, Christian Clappier, and Sonja Reidenbach for technical assistance, Ulrich Kloz and Frank van der Hoeven for generating transgenic mice, Dr. Damir Krunic (DKFZ microscopy core facility) for help with FIJI software data analysis, Dr. Manfred Ruppel (Frankfurt University) for performing scanning electron microscopy, and members of the DKFZ Laboratory Animal Facility for support. We are grateful to Gabi Frommer-Kästle for skillful help with the ultrathin sectioning, to Dr. Bernd Arnold (DFKZ Heidelberg) for providing Tie2-Cre mice, and to Dr. Hai-Kun Liu (DKFZ Heidelberg) for providing Nestin-Cre mice. This work was supported by the Deutsche Forschungsgemeinschaft (DFG, SFB-TR23 Vascular Differentiation and Remodeling) and the Chica and Heinz Schaller Foundation. A.F. is supported by the Helmholtz Society.

## Author contributions

AFe and MGA performed the majority of experiments. FT performed Western blotting and characterized the Nestin-Cre;flox-Mpdz line, IM performed immunohistochemistry, DK and TB performed CT scans and MRI, FS and AvD performed histopathological examination, IH and HW performed electron microscopy, and HI, HS, and TK provided essential material and methodology and assisted with experimental design. All authors analyzed data. AFe and AFi wrote the manuscript. AFi conceived and directed the project.

## Conflict of interest

The authors declare that they have no conflict of interest.

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
