## [Review Process File · EMBO Molecular Medicine]

Loss of Mpdz impairs ependymal cell integrity leading to perinatal-onset hydrocephalus in mice

Anja Feldner, M. Gordian Adam, Fabian Tetzlaff, Iris Moll, Dorde Komljenovic, Felix Sahn, Tobias Bäuerle, Hiroshi Ishikawa, Horst Schrotten, Thomas Korff, Ilse Hofmann, Hartwig Wolburg, Andreas von Deimling, Andreas Fischer

Corresponding author: Andreas Fischer, German Cancer Research Center DKFZ

Review timeline:

Submission date:	21 March 2016
Editorial Decision:	21 April 2016
Revision received:	21 March 2017
Editorial Decision:	06 April 2017
Revision received:	11 April 2017
Accepted:	12 April 2017

Transaction Report:

Editor: Roberto Buccione

1st Editorial Decision

21 April 2016

Thank you for the submission of your manuscript to EMBO Molecular Medicine. We have now heard back from the three Reviewers whom we asked to evaluate your manuscript.

As you will see the issues raised are important. Although I will not dwell into much detail, I would like to highlight the main points.

Firstly, while Reviewers 1 and especially 3, are more positive, Reviewer 2 is much more reserved. We identify the following fundamental issues that require your action. On one hand, all reviewers lament the insufficient quality of presentation and data. For instance, Reviewer 1 details many such instances and what should be done to remedy. The other important concern, mainly expressed by Reviewer 2, is the lack of adequate functional analysis. Reviewer 2 also notes important references to prior work were overlooked. We agree on all these points.

In conclusion, while publication of the paper cannot be considered at this stage, given the potential interest of your findings and after internal discussion, we have decided to give you the opportunity to address the above concerns.

We are thus prepared to consider a substantially revised submission, with the understanding that the Reviewers' concerns must be addressed with additional experimental data where appropriate to achieve substantial improvement of data quality, molecular analysis and an attempt at exploration of

the mechanistic consequences of MPDZ loss and that acceptance of the manuscript will entail a second round of review.

Since the required revision in this case appears to require a significant amount of time, additional work and experimentation and might be technically challenging, I would therefore understand if you chose to rather seek publication elsewhere at this stage. Should you do so and although we hope not, we would appreciate a message to this effect. Please note that it is EMBO Molecular Medicine policy to allow a single round of revision only and that, therefore, acceptance or rejection of the manuscript will depend on the completeness of your responses included in the next, final version of the manuscript.

As you know, EMBO Molecular Medicine has a "scooping protection" policy, whereby similar findings that are published by others during review or revision are not a criterion for rejection. However, I do ask you to get in touch with us after three months if you have not completed your revision, to update us on the status. Please also contact us as soon as possible if similar work is published elsewhere.

Please note that we now mandate that all corresponding authors list an ORCID digital identifier. You may do so through our web platform upon submission and the procedure takes <90 seconds to complete. We also encourage co-authors to supply an ORCID identifier, which will be linked to their name for unambiguous name identification.

I look forward to seeing a revised form of your manuscript as soon as possible.

***** Reviewer's comments *****

Referee #1 (Remarks):

Feldner et al have generated conventional and conditional knockout mice for Mpdz. Deletion of Mpdz resulted in postnatal formation of hydrocephalus due to blockage of the cerebral aqueduct. The ependymal cells of the Mpdz- deficient mice displayed defects in barrier integrity which was associated with enhanced astrogliosis and stenosis of the aqueduct. Together this study established a new mouse model for hydrocephalus by ablating the Mpdz, a gene loci that has been associated with non-syndromic hydrocephalus in humans. This mice developed a spectacular phenotype and ventricular injection experiments nicely demonstrate that this phenotype is caused by a stenosis of the cerebral aqueduct. These studies are novel, because it described the generation of the first Mpdz knockout mice. These studies are highly relevant to a broad audience and represent a powerful mouse model for hydrocephalus. The quality of the illustrations should be clearly improved as outlined below. Furthermore the mechanism of the loss of ependymal cells should be investigated in more detail. The discussion should be expanded by a discussion on alternative mechanism that could explain the observed phenotype in these mice.

My specific concerns and suggestions are listed below:

- Please provide information how many times your mice were backcrossed to C57Bl/6 background
- Please provide a survival table for Mpdz +/- heterozygous crosses and for CMVcre Mpdz mice including expected number offsprings versus number of viable offsprings and statistical analysis.
- What were incidences of neurological symptoms?
- Fig.1E: It is difficult to recognize the brain structures on the right pictures of the lower panel. I would remove them. In the last sentence of the legend of Fig.1 it is not clear to what (P4) refers to. Please clarify.
- Fig. 3: Hi labeling in figure legend is missing. The author described that the 4th ventricle is not dilated, but they do not provide histological images to support this. I would suggest to include histological sections that contain the fourth ventricle (preferentially sagittal sections including also the choroid plexus of the 4th ventricle) to illustrate that they are no difference upon deletion of

Mpdz. These images are important to provide stronger support for their conclusion of a stenosis of the cerebral aqueduct.

- Fig. 4 A Image of CD31 on Mpdz brain contains only small white dots on a black background. This picture should be removed with an appropriate high quality picture of CD31 staining. Furthermore, it is not clear how many samples/mice have been investigated.

- Fig. 4D Both histological sections contain many artifacts, most likely induced during tissue extraction and tissue preparation. High quality HE stained tissue section should be included to better demonstrate that the brain architecture is not disturbed.

- Fig. 4E. Quality of the western blot is poor. Higher quality Western blot images should be included. In addition it would be important to demonstrate that brains lack Mpdz protein expression in conventional knockout mice and in the CMV-cre Mpdz deleted mice.

- Fig. 7. Quality of the HE images is poor. Different colors, some pictures are out of focus (blurry), some pictures were taken with a dirty lens or from dirty slides. I would suggest to ask support from a pathologist to generate higher quality pictures. Furthermore high power pictures of potential dying ependymal cells would be important to determine whether cells are dying via necrosis or apoptosis. This should be combined with immunohistochemical apoptosis markers. Alternatively ependymal could be denuded because of loss of adhesion. Have you observed swimming ependymal cells in the CSF? This finding would provide information about the potential mechanism for the observed loss of the ependymal cells. In combination with GFAP staining it could also provide more insights about the sequence of events of ependymal cell loss, astrogliosis, and hydrocephalus.

- Fig. 8 Quality of GFAP staining is not optimal and difference in staining intensity is difficult to distinguish especially for the lateral ventricle sections. For the cerebral aqueduct difference is more apparent but here tissue architecture is difficult to recognize. Immunohistochemical GFAP staining works very well on paraffin embedded mouse brain sections. I would recommend to use this technique to better illustrate the intensity, distribution, and localization of the astrogliosis. Furthermore it would help to determine whether loss of ependymal cells is directly associated with astrogliosis.

- Suppl. Fig. 1 High power and high quality images of the organs should be included in order to be able to judge the tissue architecture of the organs.

- Suppl. Fig. 2. In my opinion this figure should be moved to the main figure because it provides important mechanistic insights about the potential mechanism how loss of Mpdz causes enhanced permeability. It is surprising that these experiments were performed in MCF-7 cells. These experiments should be performed in freshly isolated and cultured ependymal cells from wt and knockout mice. The approach can then also be used to determine whether deletion of Mpdz causes cell death or loss of intercellular adhesion. Acute deletion of Mpdz using the conditional knockout approach might be an efficient way to measure the impact. These experiments would provide important information about the mechanism how Mpdz loss leads to loss of ependymal cells.

- Suppl. Fig. 3. See comments for Fig. 8

- Discussion is missing alternative mechanism that could explain how loss of Mpdz causes hydrocephalus. The researchers observed a loss of ependymal cells. What could be the potential mechanism that would explain this loss.

Referee #2 (Comments on Novelty/Model System):

While the technical quality of the data presented is medium, the extent of the analysis unfortunately is insufficient. Improvement of this manuscript will necessitate a significant amount of experimentation, which exceeds what we would consider appropriate for a major revision. Notwithstanding this judgement, the research topic combines novelty and medical relevance, also the approach taken is adequate.

Referee #2 (Remarks):

In the manuscript entitled 'loss of MPDZ impairs ependymal cell integrity leading to perinatal onset hydrocephalus in mice' Feldner and colleagues describe the generation and analysis of a global and conditional allele for deletion of the junctional-associated protein MPDZ to study the mechanism of autosomal recessive, non-syndromic hydrocephalus in a mouse model. Mice with a global deletion of MPDZ developed early postnatal hydrocephalus that was associated with enlargement and thinning of the skull bones. Development of hydrocephalus was absent, when MPDZ deletion was restricted to endothelial cells. While the morphology of the choroid plexus and ependymal cells of the ventricular system appeared ultrastructurally intact, development of the hydrocephalus was associated with a progressive blockage of flow of cerebrospinal fluid through the cerebral aqueduct which was preceded by astrogliosis.

The manuscript addresses the molecular mechanisms underlying the etiology of non syndromic congenital hydrocephalus. For this purpose the authors have analyzed a constitutive as well as an inducible allele for the deletion of MPDZ (MUPP1) in the mouse, which is commendable. However, the analysis is extremely superficial and does not address a possible molecular mechanism underlying the hydrocephalus development in any detail.

While it was to be expected that the endothelial deletion of MPDZ would be without effect, the relevant cell population could have been narrowed down taking advantage of the inducible allele described in the manuscript. Nestin-Cre (expressed in radial glia cells, the embryonic precursors of ependymal cells) or FoxJ1-CreER (restricted to ependymal cells with motile cilia) would be possible driver lines promising important insights (Meletis et al. PLoS Biol 6(7): e182 doi:10.1371/journal.pbio.0060182).

In their mechanistic analysis, the authors have largely limited themselves to ultrastructure, which does not necessarily provide evidence on functional properties. A molecular analysis on e.g. tight junction composition is missing, which could have been provided by immunofluorescence staining of cryosections for relevant components. What is the distribution and localization of Jams, Pals1, Par6, PATJ? It has been shown that a mechanosensory complex in radial glia cells is important to localize PCP components and proper polarization of ependymal epithelium (Ohata et al. Pkd1 and 2 in Ventricular PCPJ. Neurosci., August 5, 2015 35(31):11153-11168). Would the absence of MPDZ influence components of the PCP pathway?

Surprisingly the authors do not comment on / reference a study that has demonstrated that loss of Myosin IXa leads to hydrocephalus development in a fashion that basically phenocopies their own findings (Abouhamed et al. Molecular Biology of the Cell Vol. 20, 5074-5085). Interestingly with Myo9a being a RhoGAP this study suggested that the loss of Myo9a resulted in elevated Rho activity causing a stimulation of Rock kinase activity, which impaired ependymal maturation. In view of the fact that in oligodendrocyte precursor cells e.g. NG2 stimulates RhoA activity and hence Rock kinase at the cell periphery via the MUPP1/Syx1 signaling pathway, there might be a direct and experimentally testable mechanistic link between MPDZ loss, localization of the polarity complexes and RhoA activity (Biname' et al. NG2 Regulates Directional Migration of OPC J. Neurosci., June 26, 2013 • 33(26):10858 -10874).

Minor points:

Introduction: „It was therefore hypothesized that disruption of the junctions between cells of the ventricular zone may be the common cause. " It had already been suggested by Al-Doradi et al. (2012) that abnormal cell-cell interactions are a common pathological mechanism for congenital hydrocephalus, which should be cited in this context.

Results page 5, "MPDZ is abundantly expressed in brain endothelial cells" This statement cannot be concluded from the denoted citations: Sitek et al. (2003): Mupp1 is abundantly expressed in the choroid plexus and seems to be localized on the apical surface of epithelial cells. Expression in epithelial cells was also described by Becamel et al (2001). Ullmer (1998) only describes the expression of Mupp1 in brain and other organs, without mentioning the cell type.

Results page 6, the authors state that CSF analysis in mice is generally not possible. However, a short literature search reveals publications on the collection and characterization of CSF in mice: Liu et al. (2008): A technique for serial collection of cerebrospinal fluid from the cisterna magna in mouse

Smith et al. (2014) Characterization of individual mouse cerebrospinal fluid proteomes
 In addition one can also find information on the composition of CSF: Cunningham et al. (2013)
 Protein changes in immunodepleted cerebrospinal fluid from a transgenic mouse model of
 Alexander disease using mass spectroscopy.

Results page 8, "The ventricular system is lined by the ependyma, a single layer of simple cuboidal to columnar epithelium with microvilli and motile cilia on the apical surface. MPDZ expression is very pronounced in this cell layer" Citation of Ullmer et al. is not relevant

Figure 4 A: PECAM1 staining of brain sections. Please provide a better quality image for MPDZ-/-
 Figure 6 B: Is the magnification indeed identical in both pictures?

Figure 8, A: SVZ in the figure but SEZ in the figure legend; B: what is indicated by the arrow?

Referee #3 (Comments on Novelty/Model System):

The study employs multiple innovative and highly relevant mouse models. The technical quality of the analyses of the mice is very high.

The study is novel - there are no reports covering the discovery described here, as far as I am aware. The medical impact is significant given that hydrocephalus is a relatively common problem affecting the nervous system.

Referee #3 (Remarks):

This study employs multiple novel genetically-engineered mouse models to demonstrate that loss of the tight junction-associated protein Mpdz leads to hydrocephalus in mice. The study flows from a relatively recent finding that the MPDZ gene locus is associated with non-syndromic hydrocephalus in humans. The study is timely, of very high quality and the conclusions are justified by the data. The use of multiple mouse genetic models is a particular strength of the manuscript. I have only a few minor points to raise:

1. The Western blot in Figure 4E is scrappy and should be improved in quality. More of the blot should be shown to indicate the specificity of the result.
2. The Discussion has a narrow focus, and could be broader. Do the findings suggest any improved strategies for treating patients who have hydrocephalus due to an MPDZ mutation?
3. What is Hi in Figure 3?
4. It looks as though "SVZ "in Figure 8 should be "SEZ" given what is in the legend.

1st Revision - authors' response

21 March 2017

Referee #1:

Feldner et al have generated conventional and conditional knockout mice for Mpdz. Deletion of Mpdz resulted in postnatal formation of hydrocephalus due to blockage of the cerebral aqueduct. The ependymal cells of the Mpdz- deficient mice displayed defects in barrier integrity which was associated with enhanced astrogliosis and stenosis of the aqueduct. Together this study established a new mouse model for hydrocephalus by ablating the Mpdz, a gene loci that has been associated with non-syndromic hydrocephalus in humans. This mice developed a spectacular phenotype and ventricular injection experiments nicely demonstrate that this phenotype is caused by a stenosis of the cerebral aqueduct. These studies are novel, because it described the generation of the first Mpdz knockout mice. These studies are highly relevant to a broad audience and represent a powerful mouse model for hydrocephalus. The quality of the illustrations should be clearly improved as outlined below. Furthermore the mechanism of the loss of ependymal cells should be investigated in more detail. The discussion should be expanded by a discussion on alternative mechanism that could explain the observed phenotype in these mice.

We thank the reviewer for the overall very positive evaluation of our work. We have addressed all of the concerns (see below) to further improve the quality of our manuscript.

My specific concerns and suggestions are listed below:

- Please provide information how many times your mice were backcrossed to C57Bl/6 background

This information is given in the Material and Method section. Mice had been backcrossed for at least nine generations.

Please provide a survival table for Mpdz +/- heterozygous crosses and for CMVcre Mpdz mice including expected number offsprings versus number of viable offsprings and statistical analysis.

We show survival of the knockout strains as s Kaplan Meier survival plots (Fig. 1E and F) and report median survival in the text (20 days and 25 days). The expected Mendelian ratios have been added to the text and we provide statistical analysis (Chi square test) of 366 offspring mice from heterozygous breedings and 167 offspring mice from Cre-flox breedings in the text.

What were incidences of neurological symptoms?

This information was already included in the main text: decreased alertness, lethargy, movement disorders, muscle weakness, and apathy.

Fig.1E: It is difficult to recognize the brain structures on the right pictures of the lower panel. I would remove them. In the last sentence of the legend of Fig.1 it is not clear to what (P4) refers to. Please clarify.

As suggested we have removed the right pictures of the lower panel. The abbreviation (P) means postnatal day. We have added this to all figure legends accordingly.

Fig. 3: Hi labeling in figure legend is missing. The author described that the 4th ventricle is not dilated, but they do not provide histological images to support this. I would suggest to include histological sections that contain the fourth ventricle (preferentially sagittal sections including also the choroid plexus of the 4th ventricle) to illustrate that they are no difference upon deletion of Mpdz. This images are important to provided stronger support for their conclusion of a stenosis of the cerebral aqueduct.

Hi, hippocampus. We have added this information. We have also added new images showing the 4th ventricle, which is not dilated (new Fig. 3).

Fig.4 A Image of CD31 on Mpdz brain contains only small white dots on a black background. This picture should be removed with an appropriate high quality picture of CD31 staining. Furthermore, it is not clear how many sample/mice have been investigated.

We apologize for this. During conversion of the file into PDF the CD31 information got lost. We now show CD31 staining and the complex vascular analysis (including endothelial-specific KO mice) in Fig. EV1. We also added the number of mice for analysis of microvessel density (n=3 mice per genotype) and the number of Tie2-Cre mice we had obtained (n>200) and which did not show any signs of hydrocephalus. This information has been added to the figure legend.

Fig.4D Both histological section contain many artifacts, most likely induced during tissue extraction and tissue preparation. High quality HE stained tissue section should be included to better demonstrate that the brain architecture is not disturbed.

We apologize for this. We had some problems of processing the brains, in particular those who were already “damaged” by the hydrocephalus. We now show several new and improved images (new Figures 3 and 5). Please note that the thickness of sections slightly differs between the developmental stages. Therefore the intensity of H&E staining also differs slightly. However, the

main message (dilation of lateral ventricles, ependymal denudation in *Mpdz*-deficient mice) can be clearly seen.

Fig 4E. Quality of the western blot is poor. Higher quality Western blot images should be included. In addition it would be important to demonstrate that brains lack *Mpdz* protein expression in conventional knockout mice and in the CMV-cre *Mpdz* deleted mice.

We agree and now show better Western blot images for both transgenic strains (new Fig. 1 and Fig. EV1).

Fig.7. Quality of the HE images is poor. Different colors, some pictures are out of focus (blurry), some pictures were taken with a dirty lens or from dirty slides. I would to suggest to ask support from a pathologist to generate higher quality pictures. Furthermore high power pictures of potential dying ependymal cells would be important to determine whether cells are dying via necrosis or apoptosis. This should be combined with immunohistochemical apoptosis markers. Alternatively ependymal could be denuded because loss of adhesion. Have you observed swimming ependymal cells in the CSF? This finding would provide information about the potential mechanism for the observed loss of the ependymal cells. In combination with GFAP staining it could also provide more insights about the sequence of events of ependymal cell loss, astrogliosis, and hydrocephalus

This figure has been improved and we provide new and more uniformly H&E staining of much better quality showing ependymal denudation in *Mpdz*-deficient mice (new Figure 5). It was impossible however to detect “dying” ependymal cells by H&E histology. We have also stained brain sections against active Caspase-3 to detect apoptotic cells. Some apoptotic cells could be detected throughout the subventricular zone at P0. At P3, were there increasing numbers of caspase-3-positive cells in the ependymal layer and the choroid plexus. At P7, when hydrocephalus was present, there were many cells positive for caspase-3 in the choroid plexus and the ependyma (Fig. 4C). Therefore it is very likely that ependymal cells are dying as a consequence of *Mpdz* loss. The H&E stainings show clearly signs of ependymal denudation (Fig. 5).

We repeated CSF analysis and found increased cell numbers in hydrocephalic *Mpdz*^{-/-} mice (compared to reference values derived in rat and human; no reliable data available for mice). However, we could not definitively proof that these cells were ependymal cells, so we do not want to put this information into the manuscript.

In Figures 3, 5 and 8 we show a time course about hydrocephalus formation, ependymal defects and astrogliosis. This suggests that both processes occur almost simultaneously. Given the new data about impaired expression of *Pals1* and increased *RhoA* activity after loss of *Mpdz* (new Figure 6 and 7), it appears quite likely that decreased barrier integrity leads to detachment of ependymal cells which is immediately followed by astrogliosis as a repair process. Unfortunately, astrogliosis blocks CSF flow through the aqueduct leading to hydrocephalus formation.

Fig.8 Quality of GFAP staining is not optimal and difference in staining intensity is difficult to distinguish especially for the lateral ventricle sections. For the cerebral aqueduct difference are more apparent but here tissue architecture is difficult to recognize. Immunohistochemical GFAP staining work very well on paraffin embedded mouse brain sections. I would recommend to use this technique to better illustrate the intensity, distribution, and localization of the astrogliosis. Furthermore it would help to determine whether loss of ependymal cells is directly associated with astrogliosis.

We thank the reviewer for this very helpful comment. We have improved the images showing GFAP staining by immunofluorescence (Fig. 8) and show in addition staining of paraffin-embedded brain sections (DAB stain, brown color) in Fig EV4.

Suppl.Fig.1 High power and high quality images of the organs should be included in order to be able to judge the tissue architecture of the organs.

We now provide high power and high quality images of liver and kidney (Fig. EV2).

Suppl. Fig2. In my opinion this figure should be moved to the main figure because it provide important mechanistic insights about the potential mechanism how loss of Mpdz causes enhance permeability. It surprising that these experiments were performed in MCF-7 cells. These experiment should be performed in freshly isolated and culture ependymal cells from wt and knockout mice. The approach can then also be used to determine whether deletion Mpdz causes cell death or loss of intercellular adhesion. Acute deletion of Mpdz using the conditional knockout approach might be efficient way to measure the impact. These experiments would provide important information about the mechanism how Mpdz loss leads to loss of ependymal cells.

We agree and now show these data in main Figure 6. There is no suitable protocol available for the isolation of ependymal cells from mice.

We adopted methods described for rats and could successfully isolate and culture mouse ependymal cells. However the cells did not proliferation in such a way that we could obtain dense cultures on transwell filters. This also precluded analyses of intercellular adhesion. To solve this problem, we have stained several adhesion proteins in mouse tissue. This revealed that loss of Mpdz leads to an almost complete loss of the interacting protein Pals1 (Figure 7). This protein belongs to the crumbs family of planar cell polarity proteins, which play a critical role for ependymal integrity and function. Loss of planar cell polarity proteins is known to cause hydrocephalus (please refer to the Discussion of our paper).

Moreover, it is known that Pals1 is linked via Mpdz to Syx, RhoA-specific GEF. Therefore we have analyzed RhoA activity in freshly isolated astrocytes from Mpdz-deficient mice (as these cells can be better cultivated in vitro). We detected a pronounced increase in RhoA activity compared to wild-type littermate controls. Increased RhoA activity is known to diminish the strength of cell-cell interactions and leads to hydrocephalus (demonstrated by the Myosin IXa knockout, Abouhamed et al, 2009).

Lastly, we employed another cell line to determine barrier integrity after Mpdz loss: human choroid plexus epithelial papilloma (HIBCPP) cells. HIBCPP cells form tight junctions, develop a high electrical resistance and minimal levels of macromolecular flux when grown on transwell filters and thereby represent an excellent model system for the blood-cerebrospinal fluid barrier (Schwerk et al, 2012). Silencing of Mpdz led to lower electrical resistance indicating impaired barrier integrity (Fig. 6).

Suppl. Fig3. See comments for fig.8

As already outlined we now also show GFAP expression using DAB staining on paraffinembedded brain section (Fig. EV4).

Discussion is missing alternative mechanism that could explain how loss of Mpdz cause hydrocephalus. The researchers observed a loss of ependymal cells. What could be the potential mechanism that would explain this loss.

We have added several points to the discussion, in particular the role of planar cell polarity proteins and RhoA activity.

Referee #2

While the technical quality of the data presented is medium, the extend of the analysis unfortunately is insufficient. Improvement of this manuscript will necessitate a significant amount of experimentation, which exceeds what we would consider appropriate for a major revision. Not withstanding this judgement, the research topic combines novelty and medical relevance, also the approach taken is adequate.

Remarks:

In the manuscript entitled 'loss of MPDZ impairs ependymal cell integrity leading to perinatal onset hydrocephalus in mice' Feldner and colleagues describe the generation and analysis of a global and conditional allele for deletion of the junctional-associated protein MPDZ to study

the mechanism of autosomal recessive, non-syndromic hydrocephalus in a mouse model. Mice with a global deletion of MPDZ developed early postnatal hydrocephalus that was associated with enlargement and thinning of the skull bones. Development of hydrocephalus was absent, when MPDZ deletion was restricted to endothelial cells. While the morphology of the choroid plexus and ependymal cells of the ventricular system appeared ultrastructurally intact, development of the hydrocephalus was associated with a progressive blockage of flow of cerebrospinal fluid through the cerebral aqueduct which was preceded by astrogliosis. The manuscript addresses the molecular mechanisms underlying the etiology of non syndromic congenital hydrocephalus. For this purpose the authors have analyzed a constitutive as well as an inducible allele for the deletion of MPDZ (MUPP1) in the mouse, which is commendable. However, the analysis is extremely superficial and does not address a possible molecular mechanism underlying the hydrocephalus development in any detail.

We highly appreciate the comments about novelty and medical relevance. Regarding the mechanistic insights we have added large data sets indicating a pivotal role of Mpdz for the proper expression of the interacting protein Pals1 in the ependymal cell layer and the control of RhoA activity. In addition we have improved the quality of many images and we even present first data about a novel mouse model: Nestin-Cre, flox-Mpdz. Please refer to details below.

While it was to be expected that the endothelial deletion of MPDZ would be without effect, the relevant cell population could have been narrowed down taking advantage of the inducible allele described in the manuscript. Nestin-Cre (expressed in radial glia cells, the embryonic precursors of ependymal cells) or FoxJ1-CreER (restricted to ependymal cells with motile cilia) would be possible driver lines promising important insights (Meletis et al. PLoS Biol 6(7): e182 doi:10.1371/journal.pbio.0060182).

We had no access to FoxJ1-Cre mice and importing them to our institution it would take much too long. However, we had the chance to cross flox-Mpdz mice with Nestin-Cre (Tronche et al, 1999). Indeed we observed the development of a hydrocephalus in Nestin-Cre^{+/+};Mpdz Δ/Δ mice which phenocopies the global knockout (Fig. EV3). As such, we provide additional evidence that the primary defect is within radial glia-derived cells, most likely ependymal cells.

In their mechanistic analysis, the authors have largely limited themselves to ultrastructure, which does not necessarily provide evidence on functional properties. A molecular analysis on e.g. tight junction composition is missing, which could have been provided by immunofluorescence staining of cryosections for relevant components. What is the distribution and localization of Jams, Pals1, Par6, PATJ? It has been shown that a mechanosensory complex in radial glia cells is important to localize PCP components and proper polarization of ependymal epithelium (Ohata et al. Pkd1 and 2 in Ventricular PCPJ. Neurosci., August 5, 2015 5(31):11153-11168). Would the absence of MPDZ influence components of the PCP pathway?

This is an excellent suggestion. Mpdz and its related protein Patj interact with components of the planar cell polarity complex. In particular the interactions with Pals1 are well studied. We detected that the expression of Pals1 in the ependyma of neonatal mice is drastically reduced (Fig. 7). However the expression of Crumbs-3 was not altered in Mpdz-deficient mice, indicating that there is not a complete loss of the planar cell polarity protein complex (Fig. 7).

In addition, we tested many antibodies to stain other tight junction and adherens junction proteins. Unfortunately, several staining approaches did not work well (JAM, Patj, Par6, Pkd1). For Occludin (expressed most abundantly in the choroid plexus) we found no changes in knockout compared to wildtype animals. For E-cadherin we also detected no significant changes (Fig. 7). Also the expression patterns of ZO1 and Claudin-5 were unremarkable in Mpdz deficient mice (Fig. S1).

Surprisingly the authors do not comment on / reference a study that has demonstrated that loss of Myosin IXa leads to hydrocephalus development in a fashion that basically phenocopies their own findings (Abouhamed et al. Molecular Biology of the Cell Vol. 20, 5074-5085). Interestingly with Myo9a being a RhoGAP this study suggested that the loss of Myo9a resulted in elevated Rho activity causing a stimulation of Rock kinase activity, which impaired ependymal maturation. In view of the fact that in oligodendrocyte precursor cells e.g. NG2

stimulates RhoA activity and hence Rock kinase at the cell periphery via the MUPP1/Syx1 signaling pathway, there might be a direct and experimentally testable mechanistic link between MPDZ loss, localization of the polarity complexes and RhoA activity (Binaeva et al. NG2 Regulates Directional Migration of OPC J. Neurosci., June 26, 2013 a^€Ä 33(26):10858 -10874).

This is again an excellent suggestion. We now discuss the Myo9a knockout in our paper. The phenotype is not entirely the same as in Mpdz^{-/-}: ventricular dilation starts earlier and several Myo9a^{-/-} mice survive to adulthood. However, both models show detachment of ependymal cells, astrogliosis and aqueductal stenosis. We also now discuss the important papers that had linked Mpp1 (Mupp1) with Syx and the activity of RhoA. To study this in Mpdz mice, we used primary astrocyte cultures and determined RhoA activity. Indeed, we detected higher levels of RhoA activity. This fits excellent to the Myosin-IXa paper in which the authors demonstrate that increased RhoA activity disturbs ependymal integrity and is causative for hydrocephalus formation. Moreover, we show that apoptosis in the ependymal cell layer occurs (Fig. 4C) and that this most likely leads to ependymal denudation (Fig. 5).

Minor points:

Introduction: a^€ZIt was therefore hypothesized that disruption of the junctions between cells of the ventricular zone may be the common cause. " It had already been suggested by Al-Doradi et al. (2012) that abnormal cell-cell interactions are a common pathological mechanism for congenital hydrocephalus, which should be cited in this context.

We have added this reference.

Results page 5, "MPDZ is abundantly expressed in brain endothelial cells" This statement cannot be concluded from the denoted citations: Sitek et al. (2003): Mupp1 is abundantly expressed in the choroid plexus and seems to be localized on the apical surface of epithelial cells. Expression in epithelial cells was also described by Becamel et al (2001). Ullmer (1998) only describes the expression of Mupp1 in brain and other organs, without mentioning the cell type.

We apologize for this mistake and therefore removed our statement. The Western blot in Fig. S1 shows that endothelial cells express Mpdz protein and we have many other data sets demonstrating the importance of Mpdz in endothelial cells during pathological angiogenesis (Tetzlaff et al, manuscript in preparation).

Results page 6, the authors state that CSF analysis in mice is generally not possible. However, a short literature search reveals publications on the collection and characterization of CSF in mice: Liu et al. (2008): A technique for serial collection of cerebrospinal fluid from the cisterna magna in mouse Smith et al. (2014) Characterization of individual mouse cerebrospinal fluid Proteomes. In addition one can also find information on the composition of CSF: Cunningham et al. (2013) Protein changes in immunodepleted cerebrospinal fluid from a transgenic mouse model of Alexander disease using mass spectroscopy.

Well, in principle it is doable however only with very skilled experience and technology, which was not available. In any case there are no suitable reference ranges for cells and metabolites defined for mouse CSF.

Results page 8, "The ventricular system is lined by the ependyma, a single layer of simple cuboidal to columnar epithelium with microvilli and motile cilia on the apical surface. MPDZ expression is very pronounced in this cell layer" Citation of Ullmer et al. is not relevant

We agree and we have removed this citation.

Figure 4 A: PECAM1 staining of brain sections. Please provide a better quality image for MPDZ^{-/-}

This unfortunately occurred during conversion of the initial file type into PDF. We now show CD31 staining in Fig. S1.

Figure 6 B: Is the magnification indeed identical in both pictures?

Yes it is.

Figure 8, A: SVZ in the figure but SEZ in the figure legend; B: what is indicated by the arrow?

We corrected this mistake. It is SVZ, subventricular zone. SEZ is often used for the same (subependymal zone. The arrow indicates strong GFAP staining.

Referee #3

The study employs multiple innovative and highly relevant mouse models. The technical quality of the analyses of the mice is very high. The study is novel - there are no reports covering the discovery described here, as far as I am aware. The medical impact is significant given that hydrocephalus is a relatively common problem affecting the nervous system.

Remarks:

This study employs multiple novel genetically-engineered mouse models to demonstrate that loss of the tight junction-associated protein Mpdz leads to hydrocephalus in mice. The study flows from a relatively recent finding that the MPDZ gene locus is associated with non-syndromic hydrocephalus in humans. The study is timely, of very high quality and the conclusions are justified by the data. The use of multiple mouse genetic models is a particular strength of the manuscript. I have only a few minor points to raise:

We thank the reviewer for the very positive evaluation of our work. We would like to mention that we added even more data sets about mechanistic insights in hydrocephalus formation (e.g. loss of Pals1, increased RhoA activity, Nestin-Cre mice) and further improved the quality of several images.

1. The Western blot in Figure 4E is scrappy and should be improved in quality. More of the blot should be shown to indicate the specificity of the result.

We have removed this Western blot and now provide new blots with a much better antibody (Fig. 1 and Fig. S1).

2. The Discussion has a narrow focus, and could be broader. Do the findings suggest any improved strategies for treating patients who have hydrocephalus due to an MPDZ mutation?

We wanted to keep the Discussion short with a narrow focus. However, we have now extended it with more emphasis on planar cell polarity, Syx-Rho signaling, comparison with the Myo9a knockout.

3. What is Hi in Figure 3?

This meant hippocampus. We have improved all figure and figure legends.

4. It looks as though "SVZ "in Figure 8 should be "SEZ" given what is in the legend.

Yes this is right. We now write uniquely SVZ (subventricular zone) in all relevant figures, figure legends and the text.

Thank you for the submission of your revised manuscript to EMBO Molecular Medicine. We have now received the enclosed reports from the referees that were asked to re-assess it. As you will see the reviewers are now globally supportive although reviewer 2 would like you to address a few remaining issues. Provided you do so carefully, I am prepared to make an editorial decision on your next, final version.

Please also deal with the following editorial amendments:

- 1) Fig 5 images appear to be close-ups of panels in Fig. 3. If this is the case, please clarify it in the figure legend
- 2) Panel P3 Mpdz^{-/-} in fig. 3 appears to be very similar to the first image in the middle row Mpdz^{-/-} in Fig. EV4. As for point 1 above, if this is the case, please mention in the figure legend.
- 3) Fig.s 3, 7, 8, EV1 and EV4 show some compression artifacts. This is not a critical issue, but if you could provide higher quality images, it would be preferable. Please also note the reviewer's concerns on the quality of figures. Although not all issues are critical, I would suggest you do your best to improve their quality.
- 4) We encourage the publication of source data, with the aim of making primary data more accessible and transparent to the reader. Would you be willing to provide a PDF file per figure that contains the original, uncropped and unprocessed scans of all or at least the key gels used in the manuscript and/or source data sets for relevant graphs? The files should be labeled with the appropriate figure/panel number, and in the case of gels, should have molecular weight markers; further annotation may be useful but is not essential. The files will be published online with the article as supplementary "Source Data" files. If you have any questions regarding this just contact me.
- 5) Every published paper includes a 'Synopsis' to further enhance discoverability. Synopses are displayed on the journal webpage and are freely accessible to all readers. They include a short description as well as 2-5 one-sentence bullet points that summarise the key NEW findings of the paper. The bullet points should be designed to be complementary to the abstract - i.e. not repeat the same text. We encourage inclusion of key acronyms and quantitative information. Please use the passive voice. Please attach this information in a separate file or send them by email, we will incorporate it accordingly. We also encourage the provision of striking image or visual abstract to illustrate your article. If you do, please provide a jpeg file 550 px-wide x 400-px high.

Please submit your revised manuscript within two weeks. I look forward to seeing a revised form of your manuscript as soon as possible.

***** Reviewer's comments *****

Referee #1 (Remarks):

The authors have clearly improved the quality of the manuscript by providing better and clearer illustrations, additional mechanistic studies and textual modification.

Referee #2 (Comments on Novelty/Model System):

The paper provides a valuable mouse model for autosomal recessive, non-syndromic hydrocephalus caused by MPDZ mutation. It offers the unique possibility of mechanistic studies of this disease.

Referee #2 (Remarks):

In the manuscript entitled 'loss of MPDZ impairs ependymal cell integrity leading to perinatal onset hydrocephalus in mice' Feldner and colleagues describe the generation and analysis of a global and

a conditional allele for the deletion of the junctional-associated protein MPDZ. The manuscript describes and characterizes these animals as a valuable mouse model for autosomal recessive, non-syndromic hydrocephalus, which allows mechanistic studies into this disease. Mice with either a global deficiency in MPDZ (MUPP1) or its loss in Nestin-positive cells, respectively, developed hydrocephalus early after birth that was associated with a blockage of flow of cerebrospinal fluid through the cerebral aqueduct.

The manuscript is a revised version that substantially benefitted from the addition of novel data in response to the comments on the original version. Commendably, the overall quality of the manuscript has improved and the model and results are more clearly presented now. In particular a possible link to the activation of RhoA has been investigated and evidence is presented that loss of MPDZ in astrocytes correlates with increased RhoA activity. A potential ameliorating effect of RhoA inhibition has not been investigated as had been described by Abouhamed et al. for Myo9a deficiency. However, the study by Abouhamed et al. has now been discussed. Also a potential role of Notch has not been investigated further. Rnd3, a Rho GTPase that leads to increased levels of Notch intracellular domain (NICD) protein, causes aqueduct ependymal cell proliferation and aqueduct stenosis as a consequence, resulting in congenital hydrocephalus (Lin, PNAS, 2013). On the positive side the polar planarity pathway protein PALS1 and tight junction structure have been investigated.

There are still specific points that require the attention of the authors.

- In their discussion the authors mention evidence for a possible compensation of cell polarity defects by INADL, but do not show these data. It would strengthen the manuscript to include these important data e.g. as supplementary information.
- The discussion could be restructured to make it more concise. Points that would deserve mentioning include:
 - A discussion of the observed astrogliosis, may be as a possible response / secondary effect to the disturbed CSF homeostasis
 - A discussion of the increased Caspase-3 activity, in the context of diminished cell-cell-junction stability
 - L1cam-deficient mice are not the only mouse model exhibiting postnatal development of hydrocephalus. Other models should also be mentioned (e.g. Myo9A, Mdnah5)
 - The quotation of the paper by Abouhamed et al. 2009 should be corrected: "However, enlargement of lateral ventricles can already be observed at E14.5 in Myo9^{-/-} embryos ..." The group describes the enlarged ventricles at postnatal stage P14.5.

Figures:

Figure 4C: Panel Mpdz^{+/+} P7: The CP is difficult to recognize? Could the authors exchange for a different more characteristic panel, possibly also with a better contrast?

Figure 5: The large picture in panel Mpdz^{-/-} at P7 appears out of focus. Can it be replaced by a better version?

Figure 8: The pictures in this panel appear heterogeneous. The counterstaining with the nuclear marker is difficult to recognize in particular in panel C. Also the contrast of the individual pictures varies greatly.

2nd Revision - authors' response

11 April 2017

First of all we would like to thank the Editor and the reviewers for their positive evaluation of our work and the helpful comments that enabled us to provide a substantially improved manuscript.

Editor:

1) Fig 5 images appear to be close-ups of panels in Fig. 3. If this is the case, please clarify it in the figure legend.

Yes this is the case and we have added this information to legend of figure 5.

2) Panel P3 Mpdz^{-/-} in fig. 3 appears to be very similar to the first image in the middle row Mpdz^{-/-} in Fig. EV4. As for point 1 above, if this is the case, please mention in the figure legend.

Yes this is the case and we have added this information to legend of figure EV4.

3) Figs 3, 7, 8, EV1 and EV4 show some compression artifacts. This is not a critical issue, but if you could provide higher quality images, it would be preferable. Please also note the reviewer's concerns on the quality of figures. Although not all issues are critical, I would suggest you do your best to improve their quality.

This might have occurred during conversion to PDF. We do not see the fancy lines at the screen but on the printed version. We now provide TIFF (300 dpi, CYMK) files for all figures. The size is pretty large, please let me know if you need another format.

4) We encourage the publication of source data, with the aim of making primary data more accessible and transparent to the reader. Would you be willing to provide a PDF file per figure that contains the original, uncropped and unprocessed scans of all or at least the key gels used in the manuscript and/or source data sets for relevant graphs? The files should be labeled with the appropriate figure/panel number, and in the case of gels, should have molecular weight markers; further annotation may be useful but is not essential. The files will be published online with the article as supplementary "Source Data" files. If you have any questions regarding this just contact me.

We now provide source data of the Western blots, the TER experiment and the RhoA gLISA. Please note that we have changed the corresponding Fig. 7E. Instead of normalization of the wild-type samples to 1, we now show the active RhoA / total RhoA ratio without normalization. The p-value has changed only marginally (0.0165 to 0.0171).

5) Every published paper includes a 'Synopsis' to further enhance discoverability...

We now provide a synopsis file.

RESPONSE TO REFEREES

Reviewer #2:

1) In their discussion the authors mention evidence for a possible compensation of cell polarity defects by INADL, but do not show these data. It would strengthen the manuscript to include these important data e.g. as supplementary information.

We now show these data in Appendix Figure S1.

2) A discussion of the observed astrogliosis, may be as a possible response / secondary effect to the disturbed CSF homeostasis.

We have added: "Ependymal damage most likely impairs the brain-CSF barrier and disturbs CSF homeostasis. We suggest that this subsequently results in the initiation of repair processes, in particular reactive astrogliosis. ... As such, this work showed that slight impairment of ependymal integrity triggers astrogliosis in the subependymal zone, resulting in aqueductal stenosis and ventriculomegaly in Mpdz^{-/-} mice."

3) A discussion of the increased Caspase-3 activity, in the context of diminished cell-cell-junction stability

We have added: "Loss of either cell-cell or cell-matrix interactions would trigger apoptosis, as observed by increased caspase-3 activity in the ependyma of Mpdz^{-/-} mice."

4) L1cam-deficient mice are not the only mouse model exhibiting postnatal development of hydrocephalus. Other models should also be mentioned (e.g. *Myo9A*, *Mdnah5*)

We had already included several other mouse models in the Discussion of the previous version. To make this clearer we changed a sentence: “Disturbance of ependymal integrity induces hydrocephalus in other mouse models, e.g. mice deficient for *Myo9a*, *aPKC-lambda*, *Numb-like*, *oligophrenin1*, *myosin-IIB*, *Dlg5*, or *Mdnah5* (Abouhamed *et al*, 2009; Imai *et al*, 2006; Kuo *et al*, 2006; Khelifaoui *et al*, 2007; Ma *et al*, 2007; Nechiporuk *et al*, 2007; Ibañez-Tallon *et al*, 2004).

5) The quotation of the paper by Abouhamed *et al*. 2009 should be corrected: "However, enlargement of lateral ventricles can already be observed at E14.5 in *Myo9*^{-/-} embryos ..." The group describes the enlarged ventricles at postnatal stage P14.5.

We thank the reviewer for this important comment. We have adopted our Discussion accordingly: “The *Myo9a* knockout model shows many similarities with *Mpdz*^{-/-} mice with distortion of the ependymal cell layer, stenosis of the aqueduct and dilation of the lateral and third ventricles. Similar as observed after silencing of *Mpdz* expression there was increased RhoA activity in cultured epithelial cells after silencing of *Myo9a* (Abouhamed *et al*, 2009).

Figure 4C: Panel *Mpdz*^{+/+} P7: The CP is difficult to recognize? Could the authors exchange for a different more characteristic panel, possibly also with a better contrast?

We enhanced contrast so that cell nuclei can be better seen.

Figure 5: The large picture in panel *Mpdz*^{-/-} at P7 appears out of focus. Can it be replaced by a better version?

We have improved this picture.

Figure 8: The pictures in this panel appear heterogeneous. The counterstaining with the nuclear marker is difficult to recognize in particular in panel C. Also the contrast of the individual pictures varies greatly.

We could not provide a panel of images in which each has exactly the same color intensity. However, we enhanced the quality of these images.

Corresponding Author Name: Andreas Fischer

Journal Submitted to: EMBo Molecular Medicine

Manuscript Number: EMM-2016-06430 R1